# Mice lacking DYRK2 exhibit congenital malformations with lung hypoplasia and altered Foxf1 expression gradient

Satomi Yogosawa [1], Makiko Ohkido[2], Takuro Horii[3], Yasumasa Okazaki [4], Jun Nakayama [5], Saishu Yoshida[1], Shinya Toyokuni [4], Izuho Hatada[3,6], Mitsuru Morimoto [7] & Kiyotsugu Yoshida [1✉]

Congenital malformations cause life-threatening diseases in pediatrics, yet the molecular mechanism of organogenesis is poorly understood. Here we show that *Dyrk2*-deficient mice display congenital malformations in multiple organs. Transcriptome analysis reveals molecular pathology of *Dyrk2*-deficient mice, particularly with respect to Foxf1 reduction. Mutant pups exhibit sudden death soon after birth due to respiratory failure. Detailed analyses of primordial lungs at the early developmental stage demonstrate that *Dyrk2* deficiency leads to altered airway branching and insufficient alveolar development. Furthermore, the Foxf1 expression gradient in mutant lung mesenchyme is disrupted, reducing Foxf1 target genes, which are necessary for proper airway and alveolar development. In ex vivo lung culture system, we rescue the expression of *Foxf1* and its target genes in *Dyrk2*-deficient lung by restoring Shh signaling activity. Taken together, we demonstrate that Dyrk2 is essential for embryogenesis and its disruption results in congenital malformation.

[1] Department of Biochemistry, The Jikei University School of Medicine, Tokyo, Japan. [2] Department of Molecular Biology, The Jikei University School of Medicine, Tokyo, Japan. [3] Laboratory of Genome Science, Biosignal Genome Resource Center, Institute for Molecular and Cellular Regulation, Gunma University, Maebashi, Gunma, Japan. [4] Department of Pathology and Biological Responses, Nagoya University Graduate School of Medicine, Nagoya, Japan. [5] Department of Life Science and Medical Bioscience, School of Advanced Science and Engineering, Waseda University, Tokyo, Japan. [6] Viral Vector Core, Gunma University Initiative for Advanced Research (GIAR), Maebashi, Gunma, Japan. [7] Laboratory for Lung Development and Regeneration, RIKEN Center for Biosystems Dynamics Research, Kobe, Japan. ✉email: kyoshida@jikei.ac.jp

Congenital malformations are a major issue in pediatric healthcare and the leading cause of infant mortality in the United States[1]. A recent study showed that an estimated 0.5 million children aged 0–59 months die from congenital anomalies[2]. The analysis of molecular pathology of congenital malformations provides a better understanding of the etiology of pediatric diseases, which also identify essential genes in normal development. Embryogenesis is a well-orchestrated process that is tightly regulated by genes related to transcription factors, morphogen gradients, and their regulators. Since congenital malformations occur during embryogenesis, these genes play important roles in multiple congenital anomalies. In addition to improving our understanding of the particular genes in development, the genetic knockout of these genes in mice often reproduces congenital malformations, providing extremely insightful information for the study of refractory pediatric diseases[3–8].

Lung development is well-orchestrated by the temporal and spatial expression of transcription factors, hormones and growth factors[6–8]. Lung morphogenesis depends on mesenchymal–epithelial interaction which is mediated by SHH, WNTs, FGFs, TGF-β and BMP4[7,9,10]. The mouse lung appears from the ventral foregut endoderm by segregating from esophagus in an embryonic day (E) 9.5 embryo. Trachea arises from the more proximal foregut tube, whereas the rest of the lung develops from two ventral buds that format the distal end of the trachea and undergoes branching morphogenesis to produce the pulmonary tree[11,12]. Many genes essential for early lung development are also required for other part of embryogenesis, and deletion of these genes sometimes leads to death in utero or neonatal lethality[13–15]. Among the transcription factors known to be crucial for lung development, the Fox family is of particular importance as a regulator. Genetic studies of mice have previously demonstrated that Foxf1 transcription in the lung mesenchyme is activated by epithelial Shh via epithelial-to-mesenchymal interaction and is required for airway branching morphogenesis[15,16]. However, the mechanisms underlying lung development have not been elucidated.

Dual-specificity tyrosine-phosphorylation-regulated kinase 2 (DYRK2) is a serine/threonine kinase that directly phosphorylates p53 at Ser46 to regulate apoptotic cell death in response to DNA damage[17–20]. The knockdown of DYRK2 increases cell proliferation in cancer cells and tumor progression[21–24]. Importantly, accumulating studies have demonstrated that DYRK2 is downregulated in various cancer tissues, and that low DYRK2 expression is closely associated with a poor prognosis[21,22,25–27]. These findings collectively indicate that DYRK2 is implicated in anti-tumor effects[20]. We recently reported that loss of Dyrk2 in mice leads to the suppression of Shh signaling to cause skeletal abnormalities[28]. However, limited information is available regarding the function of Dyrk2 during embryogenesis.

In the present study, we report the generation of Dyrk2-deficient mice using the CRISPR/Cas9 nickase system. We find that Dyrk2-deficient mice exhibit congenital malformations of multiple organs and death soon after birth due to respiratory failure. Dyrk2 is required for a gradient pattern of Foxf1 expression in the fetal lung, which is needed to coordinate airway branching morphogenesis. Collectively, we show that kinase activity of epithelial Dyrk2 is involved in proper lung mesenchymal development by regulating Shh signaling.

## Results

**Generation of *Dyrk2*-deficient mice**. We have previously shown that DYRK2 exerts anti-tumor effects in various cancer cells[21,22,24,25,29]. However, little is known about the function of Dyrk2 gene ablation during embryogenesis. To address this issue, we generated Dyrk2-deficient mice using the CRISPR/Cas9 nickase system (Supplementary Fig. 1a). Three heterozygous mice with deleted mutations (32, 19, or 34 bp deletion) in Dyrk2 gene were obtained (Supplementary Figs. 1b and 2b and Supplementary Table 2). We further intercrossed F1 heterozygous mice with three different deletion patterns to generate wild type (WT), Dyrk2[+/−], or Dyrk2[−/−] (Supplementary Fig. 1c)[30]. We then validated the loss of Dyrk2 protein expression in the corresponding tissues of E18.5 Dyrk2[−/−] embryos, while the expression levels of other Dyrk family members (Dyrk1A, 1B, and 3) remained unchanged, confirming the exclusive and precise editing of the Dyrk2 gene (Supplementary Figs. 1d, e and 2c)[30]. As shown in Supplementary Table 3, although there were no Dyrk2[−/−] homozygotes in the post-weaning pups, Dyrk2[−/−] embryos survived until E18.5, according to the Mendelian ratio. However, Dyrk2[−/−] neonates (P0) died soon after birth. These findings indicate that Dyrk2 is required for survival after birth and that it likely plays a role in embryonic organ development.

**Dyrk2-null embryos exhibit congenital malformations**. We initially confirmed that none of the three types of Dyrk2[+/−] mice showed significant defects in the size or shape of the organs (Supplementary Fig. 3). To determine the biological function of Dyrk2 during embryogenesis, we examined the gross morphology of Dyrk2[−/−] embryos for each deletion type. At E18.5, all Dyrk2[−/−] embryos displayed multiple defects, including the omphalocele phenotype, craniofacial development, short limb, and anal atresia, as well as an open eyelid phenotype at times (Fig. 1a–c and Supplementary Fig. 4a–c). In addition, abnormalities of tongue, cleft palate, and hair follicles were also observed in the mutants (Fig. 1d–f and Supplementary Fig. 4d–f). Further, limb dysmorphology was observed, including ectrodactyly, syndactyly, and polydactyly, as well as shortened radial bones (Fig. 1g, h and Supplementary Fig. 4g, h). These results suggest that Dyrk2[−/−] embryos exhibit congenital malformations in multiple organs. We hypothesized that Dyrk2 is a key gene involved in the development of the several vital organs.

To test this hypothesis, we validated the phenotypes of developmental abnormalities in Dyrk2[−/−] embryos. The Dyrk2[−/−] embryos displayed overall growth retardation. Skeletal staining revealed vertebral defects, including butterfly vertebrae, and many bone abnormalities in the ribs and radial bone in E18.5 Dyrk2[−/−] embryos (Figs. 1c, d, h and 2a and Supplementary Figs. 4c, d, h and 5a). The short arch ribs and vertebral body were also found to be poorly mineralized. Moreover, Dyrk2[−/−] embryos were found to have a severely truncated gastrointestinal tract, with shortened small and large intestines (Fig. 2b–d and Supplementary Fig. 5b–d). The mutant embryos exhibited an imperforate anus with recto-urethral fistula, anal atresia, and persistent cloaca (Fig. 2b and Supplementary Fig. 5b). These phenotypes are typical for anorectal malformations. Dyrk2 deficiency affected intestinal villus morphogenesis and proliferation patterns with omphalocele phenotypes (Fig. 2c, d and Supplementary Fig. 5c, d). The Dyrk2[−/−] embryos also displayed cardiovascular defects, although no heart defect was observed (Fig. 2e and Supplementary Fig. 5e). Defects in the left and right subclavian artery were observed in Dyrk2[−/−] embryos. Both the trachea and esophagus were hypoplastic in the Dyrk2[−/−] embryos, and the cartilaginous rings of the Dyrk2[−/−] embryo tracheas were smaller, as well as split in some cases (Fig. 2f, g and Supplementary Fig. 5f, g), suggesting a tracheal stenotic phenotype. The esophagus of the Dyrk2[−/−] embryos contained very small lumens lacking the typical folded structure. The Dyrk2[−/−] embryos also displayed aberrant patterning of renal

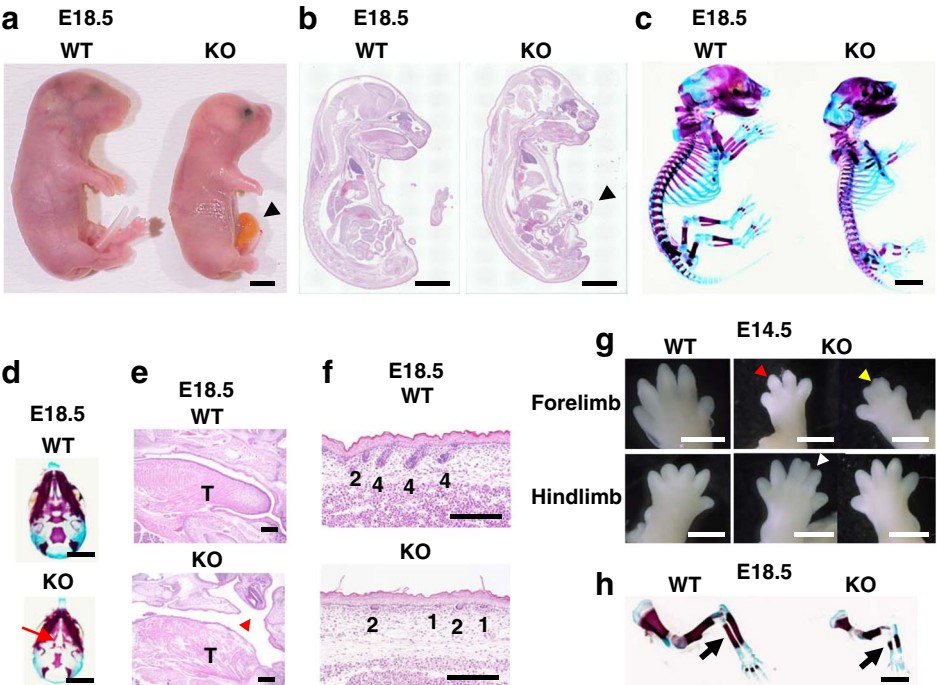

**Fig. 1 Loss of Dyrk2 leads to multiple developmental abnormalities.** *Dyrk2*$^{-/-}$ mice exhibit multiple defects in craniofacial, hair follicle, and radial/limb development. **a** Lateral views of E18.5 embryos. Arrowhead; omphalocele. **b** Lateral views of H&E sections in E18.5 embryos. Arrowhead; omphalocele. **c** Lateral views of skeleton preps in E18.5 embryos. **d** Palatal shelves in E18.5 embryos. Red arrow; cleft palates. **e** H&E staining of palate in E18.5 embryos. Red arrowhead; cleft palates. T, tongue. **f** H&E staining of skin in E18.5 embryos. Numbers denote the stages of hair follicle morphogenesis. **g** Limb dysmorphology of E14.5 embryos. The red arrowhead; ectrodactyly. The yellow arrowhead; syndactyly. The white arrowhead; syndactyly and polydactyly. **h** Radial anomalies of E18.5 embryos. Black arrows; radial bones. Scale bar: 3 mm in **a-d**, **h**, 400 μm in **e**, 200 μm in **f**, 1 mm in **g**.

medullary collecting ducts, and lobe folds, but no horseshoe kidney, which has been reported to be associated with genetic abnormality in *Shh* (Fig. 2h, i and Supplementary Fig. 5h, i). We assessed the detailed phenotypes of the respiratory organs since lung hypoplasia leads to neonatal lethality of refractory congenital disease. As expected, the deletion of *Dyrk2* caused severe lung hypoplasia and fatality from respiratory failure at P0 (see below). The *Dyrk2*$^{-/-}$ embryos exhibited lung immaturity, hypoplasia, fusion of the right lung lobes, and a large cyst on the lower left lung (Fig. 2j, k and Supplementary Fig. 5j, k). These findings collectively suggest that Dyrk2 is essential for normal lung development. The summary of abnormal phenotypes in 3 different *Dyrk2*$^{-/-}$ mice lines was shown in Supplementary Data 1. There was no significant difference among three different *Dyrk2*$^{-/-}$ mice lines (Table 1 and Fig. 2l, and Supplementary Data 1). Thus *Dyrk2*$^{-/-}$ mice exhibit developmental abnormalities and congenital malformations of multiple organs.

In contrast, the *Dyrk2*$^{-/-}$ embryos exhibited no morphological abnormalities in the brain, heart, liver, or pancreas (Supplementary Figs. 6a–e and 7a–e). A histological analysis of the stomach further revealed a thinner epithelial morphology in mutants compared to WT, although their gross morphologies were indistinguishable (Supplementary Figs. 6f and 7f). Furthermore, abnormalities of seminiferous tubule were also observed in the *Dyrk2*$^{-/-}$ embryos (Supplementary Figs. 6g and 7g).

***Dyrk2*$^{-/-}$ embryo altered expression of organogenesis associated genes.** Since *Dyrk2*$^{-/-}$ mice displayed a wide range of developmental abnormalities, we speculated that the phenotype of *Dyrk2*$^{-/-}$ mice appears at the early organogenesis stage. To understand the cause of the abnormalities in *Dyrk2*$^{-/-}$ mice, we compared gene expression profiles between the WT and *Dyrk2*$^{-/-}$ embryos at E8.5 and E10.5 using RNA collected from whole embryos. An examination of the microarray data revealed 963 individual probes in E8.5 and 733 individual probes in E10.5 with a 1.5-fold or greater change, which were selected for further analysis. We also observed that the expression levels of genes related to lymphocyte and erythrocyte development and some of top differentially expressed genes tended to increase or decrease in both E8.5 and E10.5 *Dyrk2*$^{-/-}$ embryos (Supplementary Fig. 8). The results of the GO analysis using DAVID are provided in Supplementary Table 6. Since embryogenesis is a well-orchestrated process that is tightly regulated by transcription factors, we focused on the transcription factor genes (97 probes in E8.5 and 65 probes in E10.5) that may be implicated in the abnormal phenotypes of *Dyrk2*$^{-/-}$ embryos.

The results are displayed as heatmaps (Fig. 3a, b). Among these, we focused on the downregulated genes that are reasonable for interpreting the relationship between developmental abnormalities. This comprehensive analysis revealed decreases in gene expressions associated with lung development; *Foxa2*, *Notch1*, *Foxp2*, *Nkx2.1*, and intestine development; *Cdx2*, *Foxf2*, *Foxl1*, and skeletal development; *Hoxd12*, *Hoxd13*, *Scx*, *Brachyury*, and cleft palate; *Foxf2* (Fig. 3a–c). Importantly, the expression of several Fox family genes was reduced in both E8.5 and E10.5 *Dyrk2*$^{-/-}$ embryos, suggesting that altered gene expression of Fox families may be involved in developmental abnormalities.

We validated the expression of genes that play important roles in lung development as lung defects, including hypoplasia and the fusion of the right lung lobes, have been previously found in *Dyrk2*$^{-/-}$ mice (Fig. 3d)[7]. Interestingly, *Foxf1* expression was significantly reduced in the *Dyrk2*$^{-/-}$ embryos (Fig. 3d). In mouse models, Foxf1 transcription in the lung mesenchyme is activated by epithelial Shh[15] and is required for airway branching morphogenesis[16,31]. As expected, *Shh* expression was reduced in E10.5 *Dyrk2*$^{-/-}$ embryos (Fig. 3d). Because the mutant embryos

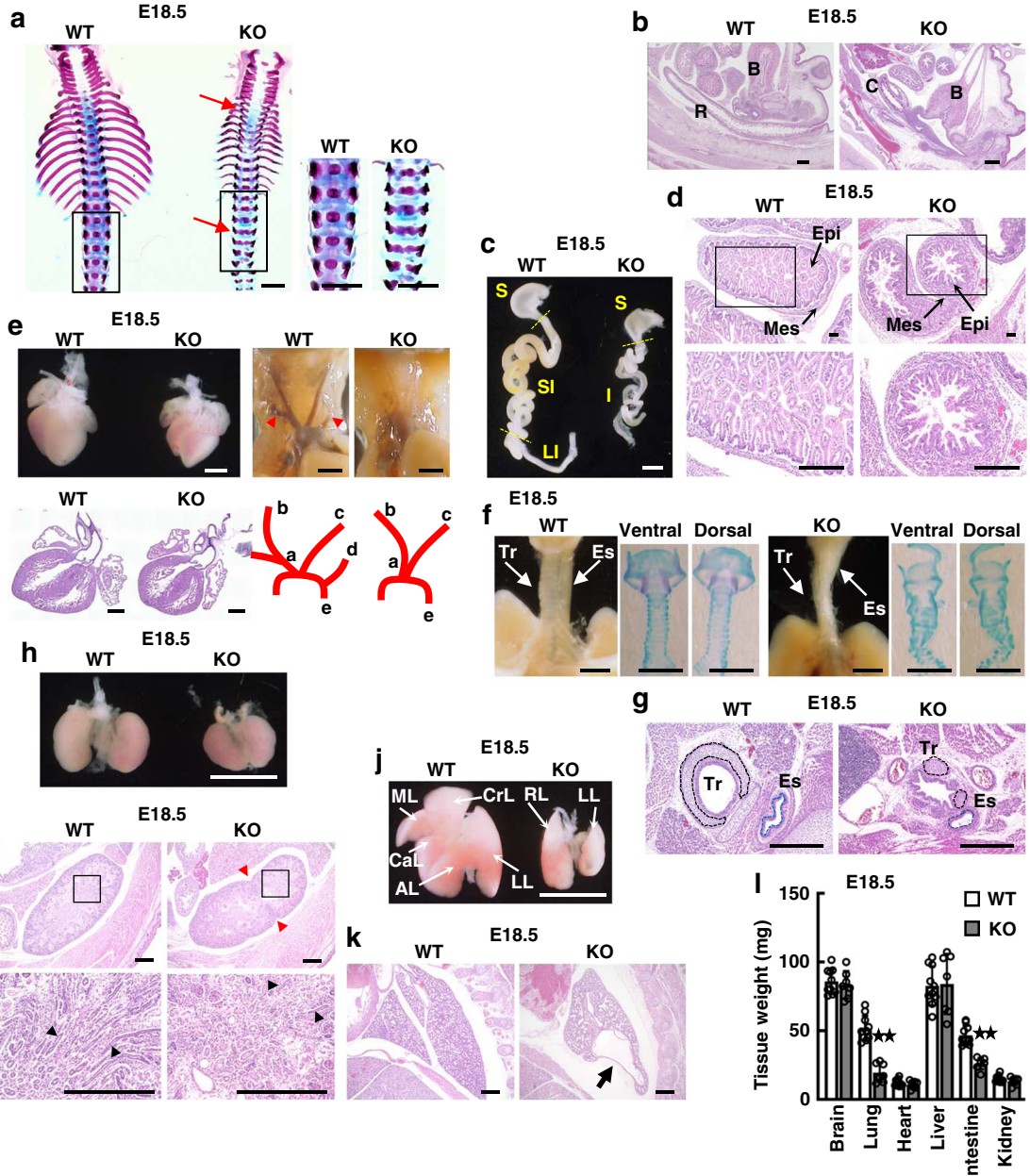

**Fig. 2 _Dyrk2_⁻/⁻ embryos exhibit congenital malformations in multiple organs.** _Dyrk2_⁻/⁻ mice exhibit vertebral, intestinal/anorectal, cardiac, trachea, esophageal, renal, and lung development. **a** Spine defects of E18.5 embryos. Red arrows; the lack of vertebral bodies and butterfly vertebrae. **b** H&E staining of the anus and cloaca in E18.5 embryos. B bladder, R rectum, C cloaca. **c** Gross morphology of intestine in E18.5 embryos. S stomach, SI small intestine, LI large intestine, I intestine. **d** H&E staining of intestine in E18.5 embryos. Epi epithelium, Mes mesenchyme. **e** Gross morphology and H&E staining of the heart and cardiac outflow tract in E18.5 embryos. Red arrowheads; the left and right subclavian arteries. Insets in each frame is a schematic of the aortic arch with the aorta and tributaries. a ascending aorta, b brachiocephalic artery, c left common carotid artery, d left subclavian artery, e descending aorta. **f** Gross morphology of trachea (Tr) and esophagus (Es), and alcian blue staining of cartilaginous rings in E18.5 embryos. **g** H&E staining of trachea (Tr) and esophagus (Es) in E18.5 embryos. Dashed black lines; cartilaginous rings. Dashed blue lines; esophagus. **h** Gross morphology of kidneys in E18.5 embryos. **i** H&E staining of kidneys in E18.5 embryos. Red arrowheads; the lobe folds. Black arrowheads; medullary collecting ducts. **j** Gross morphology of lungs in E18.5 embryos. AL accessory lobe, CaL caudal lobe, CrL cranial lobe, ML medial lobe, LL left, RL a one-lobed right lung. **k** H&E staining of lungs in E18.5 embryos. Black arrow; lung cysts. **l** Tissue weight of E18.5 embryos. Data are presented as the mean ± SD (WT: _n_ = 10; KO: _n_ = 7; **_p_ < 0.01). Scale bar: 1.5 mm (**a**), 3 mm (**c**, **h**, **j**), 400 μm (**b**, **g**, **i**, **k**), 200 μm (**d**), 1 mm (**e**, **f**).

did not lower the _Shh_ expression at E14.5, Shh may be required for initiating Foxf1 expression and may influence airway branching in primordial lung around E10.5. We further found that the primordial endoderm organs showed no significant defect in _Dyrk2_⁻/⁻ embryos at E10.5 (Fig. 3e), indicating that _Dyrk2_⁻/⁻ embryos initially exhibit genetic abnormalities around E8.5–10.5 that may be responsible for the developmental defects.

**_Dyrk2_⁻/⁻ mice die due to respiratory failure caused by upper respiratory tract malformation and lung hypoplasia.** As shown in Supplementary Table 3, _Dyrk2_⁻/⁻ mice were born in Mendelian ratios, with all _Dyrk2_⁻/⁻ neonates dying soon after birth. We found that _Dyrk2_⁻/⁻ mice died due to respiratory failure. The neonate lungs of _Dyrk2_⁻/⁻ mice contained minimal or no air and sank when placed in physiological salt solution, while those of

**Table 1 Congenital malformation phenotypes in *Dyrk2*⁻/⁻ mice.**

Craniofacial malformations (cleft palate, craniofacial abnormalities)
Hair follicle anomalies (arrested hair follicle phenotype)
Radial/Limb anomalies (shortened radial bone, hypoplasia, ectrodactyly, syndactyly, polydactyly)
Vertebral defects (lack of vertebral body, butterfly vertebrae)
Intestinal/Anorectal malformations (omphalocele phenotype, truncated gastrointestinal tract, cloaca, imperforate anus)
Cardiac defects (the left and right subclavian artery defects)
Tracheoesophageal malformations (esophageal and tracheal stenosis, smaller cartilaginous rings)
Renal malformations (hypoplasia, aberrant patterning of renal medullary collecting ducts, lobe folds)
Lung defects (hypoplasia, fusion of right lung lobes, a large cyst of lower left lung)

The summary of congenital malformation phenotypes in E18.5 *Dyrk2*⁻/⁻ embryos. Detailed abnormal phenotypes of each organ are indicated in parentheses.

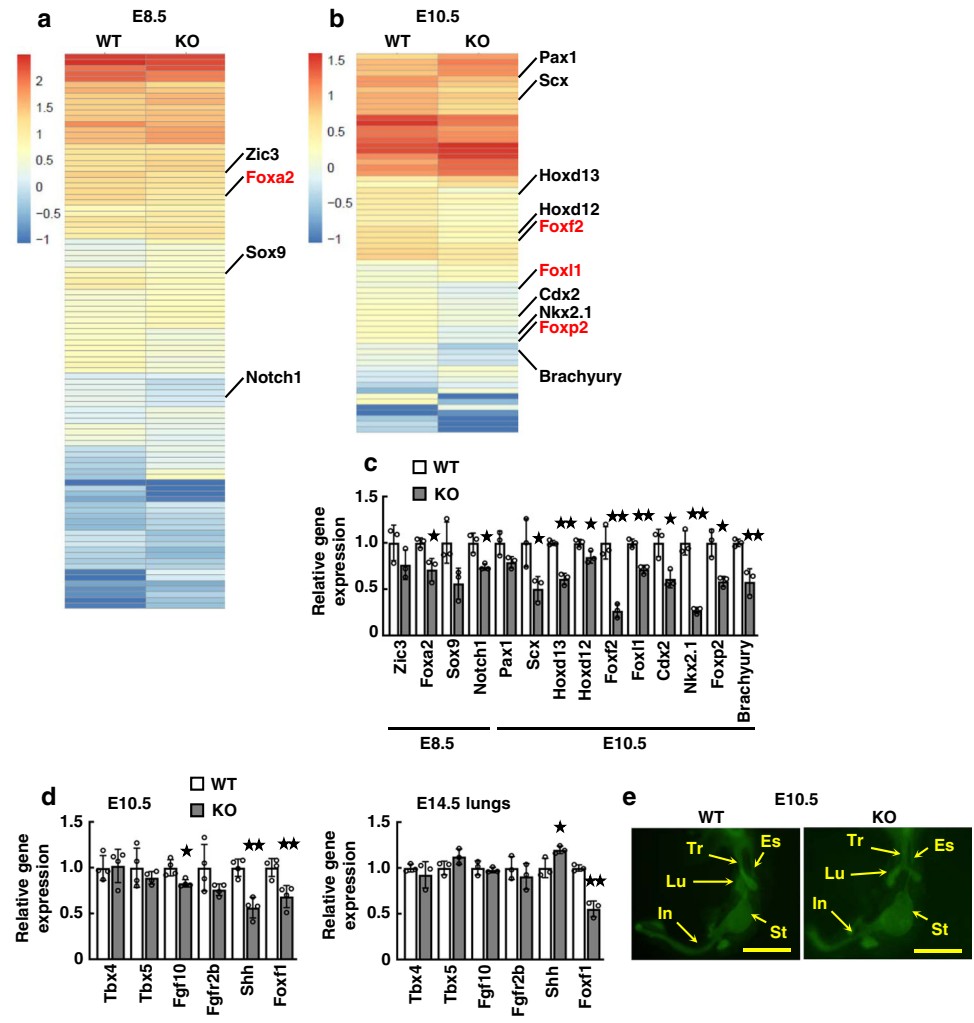

**Fig. 3 *Dyrk2*⁻/⁻ mice show reduced *Foxf1* expression.** *Dyrk2*⁻/⁻ mice display reduced expression of organogenesis associated genes. **a**, **b** Heatmap of genes related to transcription from GO analysis in E8.5 (**a**) and E10.5 (**b**) embryos. **c** Relative expression of genes related to developmental defect in E8.5 and 10.5 embryos (*n* = 3). **d** Relative expression of genes related to lung development in E10.5 embryos and E14.5 lungs (E10.5: *n* = 4; E14.5 lung: *n* = 3). **e** The representative foregut of E10.5 embryos from WT and *Dyrk2*⁻/⁻ mice with the epithelium outlined by whole-mount E-cadherin immunostaining. Tr trachea, Es esophagus, Lu lung, St stomach, In intestine. Data are presented as the mean ± SD in **c**, **d**. *$p < 0.05$; **$p < 0.01$. Scale bar: 500 μm in **e**.

WT mice floated (Fig. 4a). Micro-CT imaging analysis revealed that the mutant neonates failed to inflate their lungs (Fig. 4b and Supplementary Fig. 9a). The first breath of these pups was examined by cesarean section at E18.5. The *Dyrk2*⁻/⁻ mice failed to initiate normal breathing; however, it showed deep respiratory movements involving the whole-body muscles immediately after birth (Supplemental Movies 1 and 2). The mutant mice

subsequently became cyanotic and survived for only a few minutes. This observation indicates a cause for respiratory failure, but not neuromuscular, muscular dysfunctions, or skeletal anomalies.

We further examined the details of lung development in *Dyrk2*⁻/⁻ mice to determine the causal phenotype of lung hypoplasia and respiratory failure. At E18.5, in normal development, both alveolar epithelial cell (AEC) I and II line the

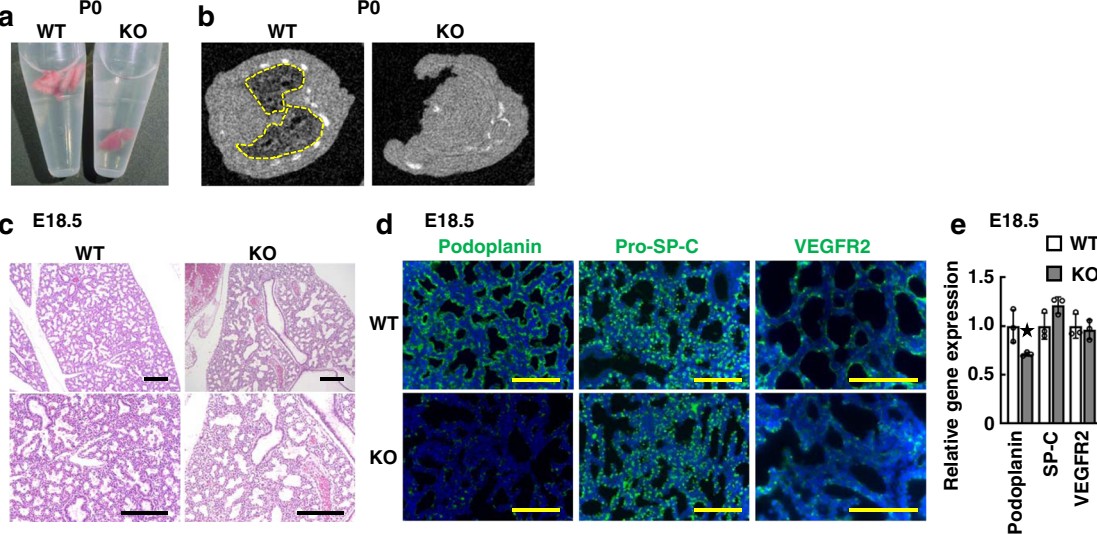

**Fig. 4 *Dyrk2*−/− mice exhibit respiratory failure due to lung hypoplasia.** *Dyrk2*−/− mice exhibit sudden death soon after birth due to respiratory failure and lung hypoplasia. **a** Neonate lungs of WT and *Dyrk2*−/− mice. **b** Micro-CT analysis of the lung from P0 WT and *Dyrk2*−/− pups. The dashed line; the area of lung inflation. **c** H&E staining of the lungs in E18.5 embryos. **d** Optical image of E18.5 lung in WT and *Dyrk2*−/− embryos. Podoplanin (green), Pro-SP-C (green), VEGFR2 (green), and DAPI (blue). **e** Relative expression of genes related to AEC I and II makers in E18.5 lungs. Data are presented as the mean ± SD ($n = 3$; *$p < 0.05$). Scale bar: 200 μm (**c**) and 100 μm (**d**).

peripheral saccules, which is the typical mature structure of the lung at this stage of gestation. As expected, normal lung inflation and histology with differentiated distal alveolar saccules were observed in the lung of WT mice (Figs. 2j, k and 4c and Supplementary Fig. 5j, k). In contrast, *Dyrk2*−/− appeared to show severe defects in the dilation with thicker septa and significantly lower weights. Furthermore, the expression of AEC I marker, Podoplanin was decreased in the E18.5 *Dyrk2*−/− lung (Fig. 4d, e) while there were no significant differences in the expression of AEC II marker, Prosurfactant Protein C (Pro-SP-C) and Surfactant Protein C (SP-C), and endothelial marker, VEGFR2. These observations indicate that the *Dyrk2*−/− embryos exhibited lung immaturity in addition to tracheal stenosis and cleft palate. Collectively, our findings suggest that upper respiratory tract malformation and lung immaturity are most likely the cause of neonatal lethality in *Dyrk2*−/− mice.

**Dyrk2 is required to form a subepithelial-to-distal expression gradient of Foxf1.** We investigated the potential role of Dyrk2 in lung hypoplasia and airway branching defects, as the fusion of the right lung lobes was observed in the *Dyrk2*−/− mutants (Fig. 2j, k). At E11.5, the primordial lung displayed the main branch, demarcating the left and right lungs, followed by several branched lung buds (Fig. 5a). The normal lung showed four tips of lung buds in the right and three tips in the left. However, the *Dyrk2*−/− lungs displayed lung buds with three on the right and two on the left. In addition, the mutants showed increased bronchial width, compared with the lung of WT mice (Fig. 5a, b). Furthermore, while the normal lung showed abundant cell proliferation and cell death in mesenchyme[7,32], the *Dyrk2*−/− lungs reduced both cell proliferation and cell death (Fig. 5c). These observations suggest that Dyrk2 is necessary for the development of airway branching.

We then sought to identify the molecular phenotype of *Dyrk2*−/− responsible for these effects on branching morphogenesis, and thus re-examined the expression reduction phenotype of Foxf1. The transcription factor Foxf1 plays an important role in epithelial–mesenchymal signaling. *Foxf1* heterozygote mutant mice have been previously found to display abnormal lung morphogenesis and a narrowing of the esophagus and trachea,

although homozygous *Foxf1*-null mice died before E10[15,16]. To better understand how Dyrk2 is involved in early lung development, we examined the expression of Dyrk2 and Foxf1. Dyrk2 was detected in epithelial cells at E11.5 and E18.5 (Fig. 5d), particularly, the subapical region of ciliated cells (FoxJ1/ Acetylated tubulin-positive cells) at late stage (Fig. 5e, f). These findings suggest that Dyrk2 express epithelial cells throughout lung development. Interestingly, a gradient expression pattern of Foxf1 protein between the subepithelial and distal mesenchyme was observed in the lungs of WT mice (Fig. 5g). Consistent with our qRT-PCR analysis (Fig. 3d), Foxf1 expression was significantly reduced in the subepithelial area of E11.5 *Dyrk2*−/− lungs, which resulted in an altered gradient expression pattern (Fig. 5g–i). Accordingly, E14.5 *Dyrk2*−/− lungs also displayed reduced expression of the Foxf1 target genes, including *αSMA*, *Myocd*, and *Hoxb7*, and increased *Wif1* expression, as described previously (Fig. 5j)[31]. To determine whether the kinase activity of Dyrk2 is required for Foxf1 expression, we conducted ex vivo embryonic lung culture with DYRK inhibitor, harmine (Fig. 5k). As expected, inhibition of the Dyrk2's kinase activity reduced Foxf1 expression. Previous reports show that in mouse models, mesenchymal Foxf1 transcription is activated by epithelial Shh and is required for airway branching morphogenesis[15,16]. To determine whether Shh activation is required for Foxf1 expression, we next conducted ex vivo *Dyrk2*−/− lung culture with Shh activator, smoothened agonist (SAG). In the *Dyrk2*−/− lungs, as expected, Shh activation restore Foxf1 expression and its targets (Fig. 5l).

These findings suggest that Dyrk2 is required to form a subepithelial-to-distal expression gradient of Foxf1 via inducing Shh signaling, which contributes to proper airway branching morphogenesis through the induction of downstream target genes.

## Discussion
In the present study, we demonstrated that the loss of *Dyrk2* represents developmental abnormalities and congenital malformations of multiple organs (Fig. 6a). We discovered that Dyrk2 is an important regulator of embryogenesis, which is

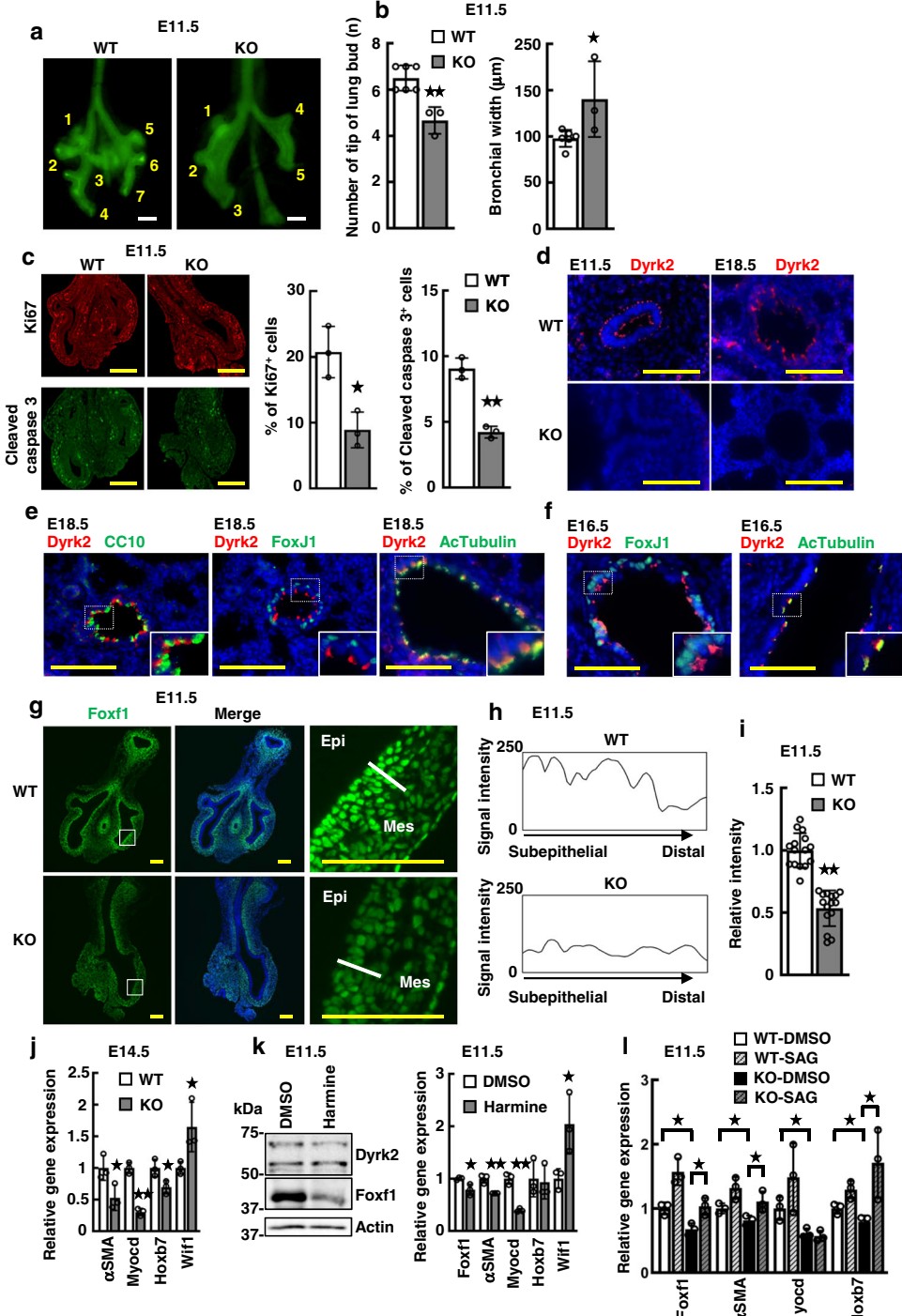

**Fig. 5 Dyrk2⁻/⁻ mice exhibit the loss of the Foxf1 expression gradient during early lung development.** Loss of Dyrk2 leads to impaired airway branching with the loss of the Foxf1 expression gradient. **a** Representative lungs from E11.5 WT and *Dyrk2⁻/⁻* mice with the epithelium outlined by whole-mount E-cadherin immunostaining. Numbers denote the tip of the lung bud. **b** Quantification of the number of tips of the lung bud and bronchial width from E11.5 WT and *Dyrk2⁻/⁻* mice. (WT: *n* = 6; KO: *n* = 3). **c** Optical image of E11.5 lung in WT and *Dyrk2⁻/⁻* embryos. Ki67 (red) and cleaved caspase 3 (green). Quantification of the number of Ki67 positive cells and cleaved caspase 3 positive cells. (*n* = 3). **d** Optical images of E11.5 and E18.5 lung in WT and *Dyrk2⁻/⁻* embryos. Dyrk2 (red) and DAPI (blue). **e, f** Optical images of E18.5 (**e**) and E16.5 (**f**) lung from WT embryos. Dyrk2 (red), CC10 (club cell marker, green), FoxJ1 (ciliated cell marker, green), Acetylated Tubulin (AcTubulin) (ciliated cell marker, green) and DAPI (blue). **g** Optical image of E11.5 lung in WT and *Dyrk2⁻/⁻* embryos. Foxf1 (green), and DAPI (blue). Epi epithelium, Mes mesenchyme. **h** A subepithelial-to-distal expression gradient of Foxf1 in E11.5 lungs. **i** Relative intensity of Foxf1 expression in the subepithelial mesenchyme (*n* = 3). **j** Relative expression of Foxf1 target genes in E14.5 lungs (*n* = 3). **k** Expression levels of Dyrk2 and Foxf1 in E11.5 lung explants treated with or without 50 μM harmine. Relative expression of *Foxf1* and its target genes in the E11.5 lung explants (*n* = 3). **l** Relative expression of *Foxf1* and its target genes in E11.5 lung explants with or without 14.8 nM smoothened agonist (SAG) (*n* = 3). Data are presented as the mean ± SD in **b**, **c**, **h–l**. *p < 0.05; **p < 0.01. Scale bar: 100 μm (**a**, **d–g**), 200 μm (**c**).

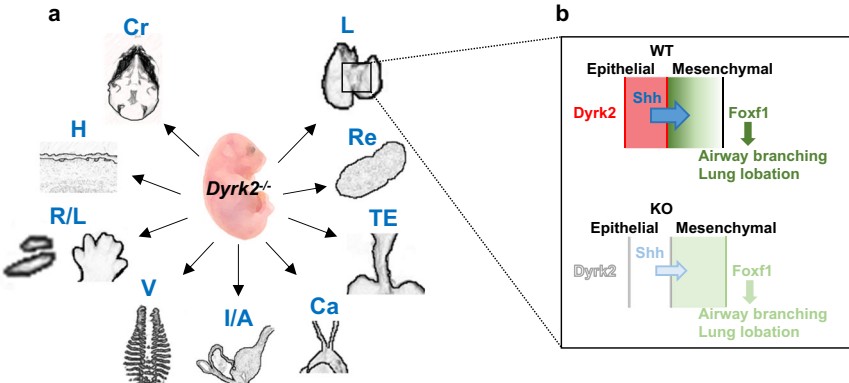

**Fig. 6 A schematic diagram of *Dyrk2* loss in mice. a** *Dyrk2*-deficient mouse mimics congenital malformation phenotypes. Cr craniofacial malformations, H hair follicle anomalies, R/L radial/limb anomalies, V vertebral defects, I/A intestinal/anorectal malformations, Ca cardiac defects, TE tracheoesophageal malformations, Re renal malformations, L lung defects. **b** Epithelial-expressed Dyrk2 is required to form a subepithelial-to-distal expression gradient of Foxf1 by regulating Shh signaling. Loss of Dyrk2 leads to the reduction of Foxf1 expression in the subepithelial area via the Shh signaling.

required for a subepithelial-to-distal expression gradient of Foxf1 via inducing Shh signaling in the primordial lung (Fig. 6b). Its disruption could be the cause of altered airway branching and alveolar development in the mutant. In this context, the *Dyrk2* gene was found to be closely related to lung development by regulating the expression pattern of Foxf1 through Shh signaling.

Based on our findings, we propose that an interaction between Dyrk2 and Shh-Foxf1 signaling is particularly important in lung development. Therefore, its disruption results in neonatal lethality due to respiratory failure. Foxf1 is a transcription factor, which is expressed in lung mesenchyme, endothelial cells, and airway smooth muscle cells[31,33]. Foxf1 promotes mesenchymal–epithelial signaling and stimulates cellular proliferation. Haploinsufficiency of Foxf1 causes severe lung malformations such as hypoplasia, fusion of right lung lobes, esophageal and tracheal stenosis, the hypoplastic tracheal cartilage, and airway branching defects[15]. Genetic studies of mice have previously demonstrated that Foxf1 acts downstream of Shh-Gli signaling via epithelial-to-mesenchymal interaction and is required for airway branching morphogenesis and lung lobation[15,16].

In the current study, our *Dyrk2*[−/−] mice exhibited significant reduction of Foxf1 expression and lung malformations such as hypoplasia, fusion of right lung lobes, esophageal, and tracheal stenosis, the hypoplastic tracheal cartilage, and airway branching defects. These defects are consistent with the phenotypes of *Foxf1*[+/−] mice, suggesting that Dyrk2 acts as a positive regulator of the Shh–Foxf1 interaction to generate the subepithelial-to-distal expression gradient of Foxf1.

Based on this finding and the recently published paper (Yoshida et al.[28]), we propose that the Dyrk2-Shh-Foxf1 axis plays a crucial role in mouse organogenesis. Given that the kinase activity of Dyrk2 is required for Foxf1 expression and the loss of *Dyrk2* results in the downregulation of Shh expression at early lung development, Dyrk2 may regulate epithelial-to-mesenchymal interaction via inducing Shh ligand expression dependent upon its kinase activity. Whereas it is still possible that other targets of Dyrk2 would be involved in the morphogenesis defects in the mutant, altered Dyrk2-Foxf1 axis is a promising pathway as a cause of the lung hypoplasia phenotype in *Dyrk2*[−/−] mice. The conditional expression of *Dyrk2* and/or *Shh* signaling and *Foxf1* in developing endodermal epithelial or splanchnic mesodermal cells may clarify this question. Further studies are required to better understand the lung development to elucidate how Dyrk2 regulates Shh and Foxf1 expression during embryogenesis.

We also showed that Dyrk2 is expressed in epithelial cells throughout the lung development, particularly in the subapical region of ciliated cells in the late embryonic lungs. The ciliated cells are the target of viral infection and play an important role in respiratory health[34]. Therefore, detailed analysis of Dyrk2 in ciliated cells could help to better understand the function of ciliated cells how to contribute to respiratory health.

We discovered that *Dyrk2*[−/−] mice exhibit defects in the multiorgan development, such as butterfly vertebrae, imperforate anus, the left and right subclavian artery defects, tracheoesophageal stenosis, shortened radial bone, polydactyly, and lung hypoplasia in addition to omphalocele, truncated gastrointestinal tract, hair follicular hypoplasia, cleft palates, and craniofacial abnormalities. *Dyrk2*[−/−] mice also exhibited decreased expression of transcription factors responsible for lung development, intestine development, skeletal development, and cleft palate. These findings indicate that Dyrk2 may play crucial roles in multi-organ development by regulating these genes.

Several knockout mice of these genes exhibit congenital malformations likewise the phenotypes of *Dyrk2*[−/−] mice. Indeed, *Foxl1*[−/−] mice have delayed villus morphogenesis, such as fewer and less defined villi[35]. *Hoxd13*[−/−] mice exhibit limb defects, such as strong reductions in length, complete absences, or improper segmentations of many metacarpal and phalangeal bones[36]. *Foxf2*[−/−] mice died with cleft palate and air-distended GI Tract within 18 h[37]. In contrast, *Foxa2*[−/−], *Notch1*[−/−], *Cdx2*[−/−], and *Brachyury*[−/−] mice show severe phenotypes that contribute to embryonic lethality during mid-gestation[38–41]. Furthermore, considering the knockout mice of Shh-Foxf1 signaling, *Shh*[−/−], *Foxf1*[−/−], and *Gli2*[−/−]*;Gli3*[−/−] mice show severe embryonic lethal phenotypes[15,42–50]. Conversely, *Gli2*[−/−], *Gli3*[−/−], *Gli2*[−/−]*;Gli3*[+/−] and *Foxf1*[+/−] mice show the phenotypes of congenital malformations mostly corresponding to those of *Dyrk2*[−/−] mice (*Gli2*[−/−]; lack of vertebral body, *Gli3*[−/−]; anal stenosis and polydactyly, *Gli2*[−/−]*;Gli3*[+/−]; agenesis of trachea and esophagus, *Foxf1*[+/−]; lung malformations and the asymmetry attachment of rib-sternum and tracheoesophageal stenosis)[15,47,50,51]. Since *Shh* and *Foxf1* expression were significantly reduced but not completely abolished in *Dyrk2*[−/−] mice, abnormal phenotypes in *Dyrk2*[−/−] mice were not severe than those in *Shh*[−/−] and *Foxf1*[−/−] mice. These findings collectively support that the Dyrk2 gene is closely related to Shh–Foxf1 signaling. In this regard, *Dyrk2*[−/−] mice provide an insight into a novel understanding of embryogenesis. Further studies are needed to better understand the embryogenesis how Dyrk2 regulates responsible genes for organogenesis.

Interestingly, patients with microdeletion/mutation of the *FOXF1* gene display multiple phenotypes, such as alveolar capillary dysplasia with misalignment of pulmonary veins (ACD/MPV), esophageal atresia with/without tracheoesophageal fistula (EA/TEF), and the VATER/VACTERL association[52–55]. In addition, mutations in *HOXD13* genes cause synpolydactyly, a limb malformation characterized by an additional digit between digits 3 and 4 and a fusion among these digits[56]. In the current study, our $Dyrk2^{-/-}$ mice exhibited multiple developmental abnormalities, such as butterfly vertebrae, imperforate anus, the left and right subclavian artery defects, tracheoesophageal stenosis, shortened radial bone, and polydactyly, and lung hypoplasia, and significant reduction of *Foxf1* and *Hoxd13* expression. These findings may suggest that DYRK2 may involve in these pediatric diseases.

Our findings also suggest that *DYRK2* is a candidate for a genetic mutation in human congenital malformation. Until now, a microdeletion in the chromosome 12q15, including the human *DYRK2* gene, or a point mutation in the gene body of *DYRK2* has never been identified in patients with human congenital malformation. In this context, an exome sequencing analysis of these patients could help to determine the relationship between the *DYRK2* gene and refractory pediatric disease.

Our results indicate that detailed analysis of the pathological and molecular phenotypes of our $Dyrk2^{-/-}$ mice may help the search for novel criteria and/or marker for the prenatal diagnosis of congenital malformation. In the future studies, the functional activation of DYRK2 during embryogenesis may have a beneficial effect in the congenital malformation.

In summary, this study demonstrated that the phenotypes of *Dyrk2*-deficient mice exhibit developmental abnormalities and congenital malformation. We confirmed that Dyrk2 is essential for survival and provide a basis for improving our understanding of embryogenesis and refractory pediatric disease.

## Methods

**Animals**. C57BL/6J and ICR mice were purchased from Charles River, Japan. All animal experiments were approved by the Animal Care and Experimentation Committee of Gunma University and Institutional Animal Care and Use Committee of Jikei University. The animals were housed in individual cages in a temperature-controlled and light-controlled environment, and had ad libitum access to chow and water.

**Generation of *Dyrk2*-deficient mice**. *Dyrk2*-deficient mice were generated using the CRISPR/Cas9 nickase system[57]. Four paired single guide RNAs (sgRNAs) (Supplementary Table 1) were designed for exons 1 and 3 of the *Dyrk2* gene and inserted into the gRNA cloning vector (Addgene). Candidate sgRNAs (Supplementary Fig. 1a) were transfected into B6 ES cells. The gene editing efficiency for the *Dyrk2* gene was confirmed using a GeneArt Genomic Cleavage Detection Kit (Thermo Fisher Scientific). Candidate 2 sgRNA was the most efficiently edited (Supplementary Fig. 2a) and was thus used as the targeting gRNA of the *Dyrk2* gene. In vitro transcribed hCas9 D10A mRNA and two sgRNAs were injected into the cytoplasm of fertilized eggs from female C57BL/6J mice. The injected embryos were transferred into the ampulla of the oviduct of pseudopregnant ICR females. A total of thirteen pups were obtained as the offspring.

**Genotyping**. To detect indel mutations of *Dyrk2*, the target site of *Dyrk2* alleles was amplified and attached with dATP, followed by cloning into the T-vector pMD20 (Takara) and DNA sequencing analysis using the BigDye Terminator v3.1 Cycle Sequencing Kit (Thermo Fisher Scientific). The primer sequences and indel mutation of the pups are listed in Supplementary Tables 1 and 2.

**Quantitative PCR (QPCR) analysis**. Total RNA was isolated from the embryos by using a RNeasy Mini Kit according to the manufacturer's instructions (Qiagen). Total RNA was synthesized using a PrimeScript™ 1st strand cDNA Synthesis Kit (Takara). Quantitative PCR was performed using the primer sequences listed in Supplementary Table 4, a KAPA SYBR FAST ABI Prism qPCR Kit (Kapa Biosystems), and PicoReal96 (Thermo Fisher Scientific), according to the manufacturer's instructions. Gene expression was normalized to that of the input control (36B4)[58].

**Western blotting**. Tissues were homogenized in buffer (10 mM Hepes, pH 7.4, 1 mM PMSF, cOmplete™ Mini Protease Inhibitor Cocktail (Sigma)) using a Milti-bead shocker (Yasui kikai) at 2500 rpm twice for 30 s. Tissue homogenates were then lysed in buffer (1% TritonX-100, 100 mM NaCl) under gentle rotation for 30 min at 4 °C and centrifuged at 14,000 rpm for 10 min. Protein concentrations were determined by DC Protein Assay (Bio-Rad). The tissue extracts (30–60 μg) were separated by SDS-PAGE and transferred to nitrocellulose membranes. The membranes were incubated with the indicated antibodies and visualized using chemiluminescence (PerkinElmer). The primary antibodies used are listed in Supplementary Table 5.

**Morphological analysis**. Whole-body and tissues from the fetal lungs were fixed in 10% neutral buffered formalin or 4% paraformaldehyde before paraffin embedding or freezing, followed by processing on regular slides. Sections were stained with hematoxylin and eosin (H&E) and the indicated antibodies. The primary antibodies used are listed in Supplementary Table 5. Images were obtained using a BZ-9000 fluorescence microscope (Keyence) and Olympus IX71 equipped with an DP73 camera. The quantification of the mean Foxf1 fluorescence intensity in the subepithelial mesenchyme of E11.5 lungs was measured using ImageJ in five places. The plot profile image obtained showed the fluorescence intensity of foxf1 along the lines of interest on the indicated images. The calculation of the Ki67 or cleaved caspase 3 positive cells in the mesenchyme of E11.5 lungs was measured using ImageJ in ten places. For whole-mount immunostaining, fixed E11.5 lungs or E10.5 embryos were stained with anti-E-Cadherin. The stained samples were cleared using Tissue-Clearing Reagent CUBIC-L and CUBIC-R+ (Tokyo Chemical Industry Co., Ltd.) and observed using fluorescence microscopy (BX51; Olympus). The tip number and width were measured using ImageJ software. The lungs of the P0 pups were assessed by micro-computerized tomography (CT) analysis (Latheta LCT-200; Hitachi). The histological analysis of the hair follicles was carried out according to morphological and histological criteria[59].

**Alcian Blue/Alizarin Red staining**. Skeletal preparations by Alcian Blue/Alizarin Red staining have also been described previously[60]. Samples were fixed in 99.5% ethanol for 10 days, placed in acetone for 1 days, and stained in 0.3% alcian blue in 70% ethanol/0.1% alizarin red in distilled water/acetic acid/70% ethanol (1:1:1:17) for 12 h. After washing with distilled water, specimens were placed in 1% KOH for 5 days and cleared by incubation in 20, 50, and 80% glycerol steps. The photos of the stained sample were taken using the digital camera (D5500; Nikon).

**Lung organ culture**. The E11.5 lungs were dissected from WT and $Dyrk2^{-/-}$ mice. The lungs were placed on a Transwell polyester membrane cell culture insert (Corning) and cultured at the air liquid interface in DMEM/ Ham's F12 medium (Nacalai tesque) supplemented with 10% FBS, penicillin-streptomycin (Nacalai tesque), and Amphotericin B (Sigma) with or without 50 μM Harmine (Tokyo Chemical Industry Co., Ltd.) or 14.8 nM smoothened agonist (SAG) (Enzo Life Sciences). DMSO was used as a diluent control. After 24 or 48 h incubation, the lung explants were collected and used for further analysis.

**Microarray analysis**. Total RNA from embryos was hybridized using a SurePrint G3 mouse GE microarray kit 8 × 60 K v3 (Agilent). The microarray data are available on the National Center for Biotechnology Information (NCBI) Gene Expression Omnibus (GEO) (accession no. GSE146614). Gene ontology (GO) analysis of the differentially expressed genes with a Z-score of over 2 or less than −2 was performed using The Database for Annotation, Visualization and Integrated Discovery (DAVID) Bioinformatics Resources 6.8. Hierarchical clustering analysis and heatmap drawing were performed using the "pheatmap" package in The Comprehensive R Archive Network with R (version 3.6.1).

**Statistics and reproducibility**. The data were analyzed with GraphPad Prism software 9.0.0 and are presented as dot plots in addition to the individual samples. Results are presented as the mean ± standard deviation (SD). Statistical significance was determined using a two-tailed Student's $t$-test. Chi-squared ($\chi^2$) analyses were performed using the online calculation chi-square tool (http://www.quantpsy.org). We repeated at least twice experiments and the exact sample size ($n$) for each experiment appear in the figure legend.

**Reporting summary**. Further information on research design is available in the Nature Research Reporting Summary linked to this article.

## Data availability

Microarray data can be accessed through the Gene Expression Omnibus (GEO) under the NCBI accession number GSE146614. Source data for all graphs in this article are included in Supplementary Data 2. Uncropped data for all blots and gels in this article are included in Supplementary Figs. 10 and 11. The information and data in this article are available from the corresponding author on request.

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

## Acknowledgements

This work was supported by grants from the Japan Society for the Promotion of Science (KAKENHI Grant Number JP 17K08672, 17H03584, 20H03519, and JP16H06276 (AdAMS)), the Jikei University Research Fund, Japan Heart Foundation Dr. Hiroshi Irisawa & Dr. Aya Irisawa Memorial Research Grant, and the Uehara Memorial Foundation. We thank Mr. Nobuaki Misawa for his excellent preparation of tissue sections; Mrs. Chikako Miura for HE staining and genotyping.

## Author contributions

S.Yogosawa and K.Y. conceived the study. S.Yogosawa and M.M. designed the project and performed experiments with the aid of M.O. Dyrk2-deficient mice were generated by T.H. and I.H. Y.O. and S.T. performed H&E staining and supported data interpretation. J.N. performed Heatmap analysis. S.Yoshida performed H&E staining. S. Yogosawa, M.O., M.M. and K.Y. wrote the manuscript with the contribution of all authors.

## Competing interests

The authors declare no competing interests.
