## [Transparent Peer Review File · Communications Biology]

Reviewers' comments:

Reviewer #1 (Remarks to the Author):

This study generated Dyrk2 null mice via CRISPR and analyzed phenotypes in multiple organs with a particular focus on the lung, and claimed a phenotypic similarity to a human disease VATER/VACTERL association. The mutant phenotypes are carefully documented, for 3 versions of the mutation, and provide new information on the role of Dyrk2. However, it is unclear how the phenotypes are connected to the biochemical activity of Dyrk2 and it is unconvincing how similar the phenotypes are to the human disease.

As pointed out in the second to the last paragraph in Discussion, the mutant mice have phenotypes not observed in the human disease. Additional evidence is needed to support the connection. Is Dyrk2 mutated or reduced in patients? With the current data, the disease connection is a speculation for Discussion, instead of the main conclusion.

Although expression microarrays were done on whole embryos, they narrowly focused on VATER/VACTERL-related genes, a few selected genes of lung development, and transcription factors. It's unclear if those genes were selected to make the disease connection. What are the other top differentially expressed genes? Also, how many biological replicates are there for microarray? Which organs and cell types express Dyrk2?

Minor:

Fig. 4d: why does the bottom row have fewer genes than the top row?

Fig. S5d: the mutant pancreas looks different.

Table S4: Mouse gene names only capitalize the first letter.

Line 110: other DYRK families should be "family members"?

Line 132: "To prove this hypothesis". In biomedical research, one does not "prove" but "test".

Fig. 2b: Please clarify "radial anomalies".

Fig. 5 caption: "through" implies causality, which is not supported by the current data.

Reviewer #2 (Remarks to the Author):

The authors present an interesting and novel study of dual-specificity tyrosine-phosphorylation-regulated kinase 2 (Dyrk2) during embryonic development in mice. Although DYRK2 function has been thoroughly studied in cancer biology, including work published by this group, the role of DYRK2 during embryonic development has been described in drosophila, c. elegans, and zebrafish, but not in mammals. Using CRISPR/Cas-9 gene editing, the authors generated 3 Dyrk2 deletion mouse lines. They report that Dyrk2 homozygous mutant mice die after birth with multiple congenital malformations including malformations that are commonly reported in human patients with VATER or VACTERL association. Although genetic analysis has been reported in human patients with VACTERL association, there is not a clear genetic mechanism that has been identified as being responsible for the disease. Furthermore, several mouse models of VACTERL have been generated; however, the phenotype of these models is often different from the disease observed in humans. Thus, identification of the genetic and developmental mechanisms responsible for VACTERL association is an important objective. In addition, although there are clear criteria for making the diagnosis of VACTERL association in patients, many patients present with additional congenital anomalies that are not typically included in the spectrum of this disease association namely lung hypoplasia which is a focus of this study. Better understanding of the genetic basis for the observed heterogeneity in patients is another important goal.

The authors present that several congenital malformations were present at birth in Dyrk2 mutant mice including the following common features of VACTERL association: vertebral anomalies, anal atresia,

cardiovascular defects (although not the typical defects in heart development), tracheal defects (tracheoesophageal fistula), esophageal defects (esophageal atresia), renal defects, and limb malformations. Additional malformations that are not typical in patients with VACTERL but that were found in *Dyrk2* mutant mice include craniofacial defects with cleft palate, lung hypoplasia and cystic lung malformations, omphalocele, and skin defects. The authors concluded that death of *Dyrk2* mutant mice occurs after birth and is due to lung hypoplasia. The authors also identified several changes in gene expression including genes associated with lung development and VACTERL association. Because the lung hypoplasia is believed to be the cause of lethality in this mouse model, defects in lung development were analyzed with both defects in branching morphogenesis and septation being reported. The authors finally investigated the molecular mechanism that might be responsible for lung hypoplasia and demonstrate that loss of *DYRK2* function is associated with decreased *Foxf1* expression and loss of the normal subepithelial gradient of *FOXF1* protein. Importantly, variants in *FOXF1* has been implicated in human VACTERL and *FOXF1* is known to play an important role in lung development.

Despite these interesting findings and the need to better understand the genetic and developmental mechanisms responsible for VACTERL, there are several concerns that could be addressed to improve the strength of the authors' conclusions.

1. Although the authors clearly demonstrate that *DYRK2* is required for embryonic development of multiple organs, it is unclear where (in what organs and in which cell types) and at what time points during development *DYRK2* is acting. Before presenting the *Dyrk2* mutant mouse data it would be helpful to show where *Dyrk2* is expressed either by in situ or immunofluorescence staining.
2. The link between human VACTERL association and *Dyrk2* loss of function is a central point of this manuscript; however, this link is not clear. Microarray analysis and exome sequencing have both been conducted in patients with VACTERL; however, neither copy number variants including the *DYRK2* locus nor likely pathogenic variants in the *DYRK2* gene have been identified. Furthermore, the phenotype that the *Dyrk2* mutant mice present with includes several defects found in patients with VACTERL association; however, the defects are more severe and include additional anomalies not commonly found in patients with VACTERL. These data suggest that *DYRK2* plays an important role in multiple aspects of development but not necessarily in the mechanisms responsible for VACTERL disease in humans. Decreasing the emphasis on the link with VACTERL and putting more of a focus on the organs affected by *Dyrk2* deletion, and especially the mechanisms responsible for these defects, would be more appropriate.
3. Craniofacial malformations, including cleft palate, are not common in patients with VACTERL; however, 4-5% of patients with esophageal atresia/tracheoesophageal fistula are known to have cleft palate. Some clarifying discussion about this association is warranted. It is also not clear if the *Dyrk2* deletion mice have tracheoesophageal fistula and esophageal atresia (TEF/EA). There appear to be defects in tracheal development and in the esophagus but is there TEF/EA?
4. The finding of cleft palate in *Dyrk2* mutant mice also makes it unclear what the cause of neonatal lethality is in these animals. The lungs of the *Dyrk2* mutant mice appear underdeveloped and have cystic malformation of the left lung lobe; however, these defects should not impair their ability to inflate if the pups try to breathe after birth and if the airway is not obstructed. Mice with craniofacial defects, in particular with cleft palate, often die after birth due to malpositioning of the tongue and occlusion of the airway. This results in the phenotype observed the *Dyrk2* mutant mice where the newborn pups attempt to breathe but the lungs fail to inflate as demonstrated by the "sinking lung phenotype". Either lung/airway specific deletion of *Dyrk2* or bypassing the craniofacial defects via tracheostomy would help to determine if the airway/lung defects or the craniofacial malformation is responsible for neonatal lethality.
5. The phenotype of the *Dyrk2* mice is very interesting but warrants more careful analysis. One

general concern is how penetrant the individual defects are? Do all of the mutant mice have the same pattern of defects or is there some variability in the anomalies in different animals. Also, is there any difference between the 3 different Dyrk2 mutant lines? The authors show that there is complete loss of DYRK2 protein in all 3 lines. The skeletal defects are not well visualized with micro-CT and might be better demonstrated using skeleton preps with Alcian blue and Alizarin red staining where the entire skeleton can be more easily visualized and measured. This would help to better characterize and demonstrate the craniofacial, vertebral, and limb defects. Alcian blue staining of the trachea would help to better characterize and demonstrate defects in tracheal cartilage development. Similarly, the defects in kidney development are also not clearly demonstrated. What are the structures in the Dyrk2 mutant kidneys that are not normally developed?

6. Although the authors do not investigate or speculate on the cellular mechanisms responsible for the multiple anomalies in the Dyrk2 mutant mice, their background studying Dyrk2 regulation of cell cycling in cancer biology would suggest a hypothesis where loss of Dyrk2 function during development might cause increased or inappropriate cell proliferation and loss of cellular differentiation. Instead the authors report that Dyrk2 mutant mice show hypoplasia of multiple structures. Could the authors demonstrate or speculate on the mechanisms responsible for the generalized hypoplasia found in multiple organs in the Dyrk2 mutant mice and how this differs from the role of Dyrk2 in regulation of cell cycling and tumor formation?

7. To understand the genetic mechanisms responsible for the malformations found in Dyrk2 mutant mice, the authors conducted gene expression array analysis and qRT-PCR experiments to identify changes in gene expression associated with these developmental anomalies. The selection of the developmental timepoints at which these gene expression experiments were conducted was based on the hypothesis that differences would be present during "early organogenesis" at embryonic day 8.5 and 10.5; however, it is unclear if any morphological defects were evident in Dyrk2 mutant mice at these time points. Dissection and whole mount or histological analysis of Dyrk2 mutant embryos would help to determine when defects in development are first evident. These experiments would also help to better determine at what developmental timepoint(s) experiments to identify changes in gene expression should be conducted to identify expression changes that might be responsible for the observed developmental defects.

8. Given the spectrum malformations identified in the Dyrk2 mutant mice, it is not surprising that the authors found multiple gene expression changes in transcription factor genes. The authors focused on changes in gene expression where the genes of interest were decreased in expression; however, increases in gene expression are also likely to cause abnormal development. Given the authors background with Dyrk2, any evidence or speculation about how loss of DYRK2 function might directly affect gene expression would help to better understand the genetic mechanisms responsible for the genetic and developmental abnormalities observed in the Dyrk2 mutant mice. Furthermore, a more complete analysis of the identified gene expression changes would be helpful as well as eventual depositing of the gene expression array results in a transcriptome analysis database such as the gene expression omnibus (GEO).

9. The authors focus on the developmental defects associated with lung hypoplasia due to their hypothesis that lung hypoplasia is responsible for neonatal lethality in Dyrk2 mutant mice. Deletion of Dyrk2 appears to impact both branching morphogenesis and septation of the gas exchange regions of the lung. It would be helpful for the authors to show where (in which cell populations) and when during development Dyrk2 expression is evident in the airway and lungs. Furthermore, because of the focus on lung hypoplasia, a more thorough demonstration of when these defects in lung development become evident and whether Dyrk2 is affecting cell cycling is important and worthwhile.

10. Finally, the authors conclude that deletion of Dyrk2 has a direct or possibly secondary impact on Foxf1 gene expression and regulation of a subepithelial protein gradient. Although Foxf1 is clearly an interesting candidate gene given its implication in human VACTERL association as well as in lung

development, it is not clear how loss of DYRK2 function causes a change in Foxf1 expression or if this is one among many developmentally important genes that are impacted by Dyrk2 deletion. Rescuing the lung phenotype with restoration of FOXF1 function would demonstrate the unique importance of this finding.

Reviewer #3 (Remarks to the Author):

In this manuscript, Yogosawa et al. used CRISPR/Cas9 genome editing to generate three distinct Dyrk2^{-/-} mouse lines and demonstrate that disruption of Dyrk2 gene in mice resembles a severe developmental human disorder VATER/VACTERL. They found that Dyrk2^{-/-} mice exhibit in reduction of several genes that are linked to VATER/VACTERL in humans. Among these genes was the FOXF1 transcription factor, which regulates epithelial-mesenchymal signaling in the developing lung and other organs. The authors propose that DYRK2 regulates lung development via FOXF1, and that FOXF1 deficiency is responsible for lung hypoplasia in Dyrk2^{-/-} mice. Overall, the experiments are well performed and all three Dyrk2^{-/-} mouse lines characterized very rigorously. There is a convincing evidence that Dyrk2^{-/-} mice have all phenotypes typical for VATER/VACTERL, including morphological, histological and gene expression changes. The manuscript is well written and easy to follow. Dyrk2^{-/-} mice will be useful to study pathogenesis of VATER/VACTERL. Several changes are suggested to improve quality of the manuscript:

Major comments:

1. In Fig. 2G, it is unclear whether cartilage rings are fragmented in Dyrk2^{-/-} mice. This can be an artefact of the sectioning. The cartilage rings should be stained, and the whole trachea should be shown from both dorsal and ventral sides to examine cartilage rings.
2. The authors should remove statements from the Results and Discussion sections regarding a possibility of "unrecognized phenotypes in VATER/VACTERL patients". These statements are speculative even for the Discussion section. Human defects in VATER/VACTERL are extensively characterized using imaging studies, autopsies and histological specimens.
3. Mechanisms of respiratory insufficiency in Dyrk2^{-/-} mice remain unclear since the lung has fairly normal histological structure (except a single cyst) (Figs 2J and 5C). FOXF1 deficiency is linked to the loss of alveolar capillaries (alveolar capillary dysplasia) in mice and humans (see a recent reference: PMID 31199666). The authors should stain Dyrk2^{-/-} lungs for endothelial markers, such as PECAM-1, to determine if the alveolar capillaries are missing or misplaced in the alveolar region. This will explain the respiratory insufficiency in Dyrk2^{-/-} mice.
4. It will be also helpful to stain Dyrk2^{-/-} lungs for endothelial progenitor cells (cKIT+PECAM1+CD45-) that are precursors of capillary endothelial cells. These cells are dependent on FOXF1 (ref, PMID: 31233341) and their loss can explain respiratory insufficiency. FACS is an alternative option to determine the number of cKIT+ endothelial progenitor cells.
5. There are no data to show the lung "immaturity" in Dyrk2^{-/-} mice (line 155). This will require the immunostaining for SP-C (type 2 marker) and T1a or Aquaporin 5 (type 1 markers) as well as qRT-PCR for surfactant-associated proteins.
6. It will be helpful to add proliferation/apoptosis studies to identify possible mechanisms of lung hypoplasia in Dyrk2^{-/-} mice.
7. Mechanistically, it is unclear whether DYRK2 can regulate FOXF1. Are these genes expressed in the same cells during embryogenesis? Does DYRK2 have a gradient of expression in the lung mesenchyme, which is similar to the FOXF1 gradient?

Minor comments:

1. Main Figure 1 can be moved to Suppl. Materials.
2. Line 154: figure 5 is presented out of order and between figures 2 and 3.
3. It will be helpful to provide DYRK2 expression pattern during embryonic development.

Reviewers' comments:

Reviewer #1 (Remarks to the Author):

This study generated Dyrk2 null mice via CRISPR and analyzed phenotypes in multiple organs with a particular focus on the lung, and claimed a phenotypic similarity to a human disease VATER/VACTERL association. The mutant phenotypes are carefully documented, for 3 versions of the mutation, and provide new information on the role of Dyrk2. However, it is unclear how the phenotypes are connected to the biochemical activity of Dyrk2 and it is unconvincing how similar the phenotypes are to the human disease. As pointed out in the second to the last paragraph in Discussion, the mutant mice have phenotypes not observed in the human disease. Additional evidence is needed to support the connection. Is Dyrk2 mutated or reduced in patients? With the current data, the disease connection is a speculation for Discussion, instead of the main conclusion. Although expression microarrays were done on whole embryos, they narrowly focused on VATER/VACTERL-related genes, a few selected genes of lung development, and transcription factors. It's unclear if those genes were selected to make the disease connection. What are the other top differentially expressed genes? Also, how many biological replicates are there for microarray? Which organs and cell types express Dyrk2?

Minor:

Fig. 4d: why does the bottom row have fewer genes than the top row?

Fig. S5d: the mutant pancreas looks different.

Table S4: Mouse gene names only capitalize the first letter.

Line 110: other DYRK families should be "family members"?

Line 132: "To prove this hypothesis". In biomedical research, one does not "prove" but "test".

Fig. 2b: Please clarify "radial anomalies".

Fig. 5 caption: "through" implies causality, which is not supported by the current data.

Reviewer #2 (Remarks to the Author):

The authors present an interesting and novel study of dual-specificity tyrosine-

phosphorylation-regulated kinase 2 (Dyrk2) during embryonic development in mice. Although DYRK2 function has been thoroughly studied in cancer biology, including work published by this group, the role of DYRK2 during embryonic development has been described in drosophila, c. elegans, and zebrafish, but not in mammals. Using CRISPR/Cas-9 gene editing, the authors generated 3 Dyrk2 deletion mouse lines. They report that Dyrk2 homozygous mutant mice die after birth with multiple congenital malformations including malformations that are commonly reported in human patients with VATER or VACTERL association. Although genetic analysis has been reported in human patients with VACTERL association, there is not a clear genetic mechanism that has been identified as being responsible for the disease. Furthermore, several mouse models of VACTERL have been generated; however, the phenotype of these models is often different from the disease observed in humans. Thus, identification of the genetic and developmental mechanisms responsible for VACTERL association is an important objective. In addition, although there are clear criteria for making the diagnosis of VACTERL association in patients, many patients present with additional congenital anomalies that are not typically included in the spectrum of this disease association namely lung hypoplasia which is a focus of this study. Better understanding of the genetic basis for the observed heterogeneity in patients is another important goal.

The authors present that several congenital malformations were present at birth in Dyrk2 mutant mice including the following common features of VACTERL association: vertebral anomalies, anal atresia, cardiovascular defects (although not the typical defects in heart development), tracheal defects (tracheoesophageal fistula), esophageal defects (esophageal atresia), renal defects, and limb malformations. Additional malformations that are not typical in patients with VACTERL but that were found in Dyrk2 mutant mice include craniofacial defects with cleft palate, lung hypoplasia and cystic lung malformations, omphalocele, and skin defects. The authors concluded that death of Dyrk2 mutant mice occurs after birth and is due to lung hypoplasia. The authors also identified several changes in gene expression including genes associated with lung development and VACTERL association. Because the lung hypoplasia is believed to be the cause of lethality in this mouse model, defects in lung development were analyzed with both defects in branching morphogenesis and septation being reported. The authors finally investigated the molecular mechanism that might be responsible for lung hypoplasia and demonstrate that loss of DYRK2 function is associated with decreased Foxf1 expression and loss of the normal subepithelial gradient of FOXF1 protein. Importantly, variants in FOXF1 has been implicated in human VACTERL and FOXF1 is known to play an

important role in lung development.

Despite these interesting findings and the need to better understand the genetic and developmental mechanisms responsible for VACTERL, there are several concerns that could be addressed to improve the strength of the authors' conclusions.

1. Although the authors clearly demonstrate that DYRK2 is required for embryonic development of multiple organs, it is unclear where (in what organs and in which cell types) and at what time points during development DYRK2 is acting. Before presenting the Dyrk2 mutant mouse data it would be helpful to show where Dyrk2 is expressed either by in situ or immunofluorescence staining.

2. The link between human VACTERL association and Dyrk2 loss of function is a central point of this manuscript; however, this link is not clear. Microarray analysis and exome sequencing have both been conducted in patients with VACTERL; however, neither copy number variants including the DYRK2 locus nor likely pathogenic variants in the DYRK2 gene have been identified. Furthermore, the phenotype that the Dyrk2 mutant mice present with includes several defects found in patients with VACTERL association; however, the defects are more severe and include additional anomalies not commonly found in patients with VACTERL. These data suggest that DYRK2 plays an important role in multiple aspects of development but not necessarily in the mechanisms responsible for VACTERL disease in humans. Decreasing the emphasis on the link with VACTERL and putting more of a focus on the organs affected by Dyrk2 deletion, and especially the mechanisms responsible for these defects, would be more appropriate.

3. Craniofacial malformations, including cleft palate, are not common in patients with VACTERL; however, 4-5% of patients with esophageal atresia/tracheoesophageal fistula are known to have cleft palate. Some clarifying discussion about this association is warranted. It is also not clear if the Dyrk2 deletion mice have tracheoesophageal fistula and esophageal atresia (TEF/EA). There appear to be defects in tracheal development and in the esophagus but is there TEF/EA?

4. The finding of cleft palate in Dyrk2 mutant mice also makes it unclear what the cause of neonatal lethality is in these animals. The lungs of the Dyrk2 mutant mice appear underdeveloped and have cystic malformation of the left lung lobe; however, these defects should not impair their ability to inflate if the pups try to breathe after birth and if

the airway is not obstructed. Mice with craniofacial defects, in particular with cleft palate, often die after birth due to malpositioning of the tongue and occlusion of the airway. This results in the phenotype observed the Dyrk2 mutant mice where the newborn pups attempt to breathe but the lungs fail to inflate as demonstrated by the “sinking lung phenotype”. Either lung/airway specific deletion of Dyrk2 or bypassing the craniofacial defects via tracheostomy would help to determine if the airway/lung defects or the craniofacial malformation is responsible for neonatal lethality.

5. The phenotype of the Dyrk2 mice is very interesting but warrants more careful analysis. One general concern is how penetrant the individual defects are? Do all of the mutant mice have the same pattern of defects or is there some variability in the anomalies in different animals. Also, is there any difference between the 3 different Dyrk2 mutant lines? The authors show that there is complete loss of DYRK2 protein in all 3 lines. The skeletal defects are not well visualized with micro-CT and might be better demonstrated using skeleton preps with Alcian blue and Alizarin red staining where the entire skeleton can be more easily visualized and measured. This would help to better characterize and demonstrate the craniofacial, vertebral, and limb defects. Alcian blue staining of the trachea would help to better characterize and demonstrate defects in tracheal cartilage development. Similarly, the defects in kidney development are also not clearly demonstrated. What are the structures in the Dyrk2 mutant kidneys that are not normally developed?

6. Although the authors do not investigate or speculate on the cellular mechanisms responsible for the multiple anomalies in the Dyrk2 mutant mice, their background studying Dyrk2 regulation of cell cycling in cancer biology would suggest a hypothesis where loss of Dyrk2 function during development might cause increased or inappropriate cell proliferation and loss of cellular differentiation. Instead the authors report that Dyrk2 mutant mice show hypoplasia of multiple structures. Could the authors demonstrate or speculate on the mechanisms responsible for the generalized hypoplasia found in multiple organs in the Dyrk2 mutant mice and how this differs from the role of Dyrk2 in regulation of cell cycling and tumor formation?

7. To understand the genetic mechanisms responsible for the malformations found in Dyrk2 mutant mice, the authors conducted gene expression array analysis and qRT-PCR experiments to identify changes in gene expression associated with these developmental anomalies. The selection of the developmental timepoints at which these gene expression

experiments were conducted was based on the hypothesis that differences would be present during “early organogenesis” at embryonic day 8.5 and 10.5; however, it is unclear if any morphological defects were evident in *Dyrk2* mutant mice at these time points. Dissection and whole mount or histological analysis of *Dyrk2* mutant embryos would help to determine when defects in development are first evident. These experiments would also help to better determine at what developmental timepoint(s) experiments to identify changes in gene expression should be conducted to identify expression changes that might be responsible for the observed developmental defects.

8. Given the spectrum malformations identified in the *Dyrk2* mutant mice, it is not surprising that the authors found multiple gene expression changes in transcription factor genes. The authors focused on changes in gene expression where the genes of interest were decreased in expression; however, increases in gene expression are also likely to cause abnormal development. Given the authors background with *Dyrk2*, any evidence or speculation about how loss of *DYRK2* function might directly affect gene expression would help to better understand the genetic mechanisms responsible for the genetic and developmental abnormalities observed in the *Dyrk2* mutant mice. Furthermore, a more complete analysis of the identified gene expression changes would be helpful as well as eventual depositing of the gene expression array results in a transcriptome analysis database such as the gene expression omnibus (GEO).

9. The authors focus on the developmental defects associated with lung hypoplasia due to their hypothesis that lung hypoplasia is responsible for neonatal lethality in *Dyrk2* mutant mice. Deletion of *Dyrk2* appears to impact both branching morphogenesis and septation of the gas exchange regions of the lung. It would be helpful for the authors to show where (in which cell populations) and when during development *Dyrk2* expression is evident in the airway and lungs. Furthermore, because of the focus on lung hypoplasia, a more thorough demonstration of when these defects in lung development become evident and whether *Dyrk2* is affecting cell cycling is important and worthwhile.

10. Finally, the authors conclude that deletion of *Dyrk2* has a direct or possibly secondary impact on *Foxf1* gene expression and regulation of a subepithelial protein gradient. Although *Foxf1* is clearly an interesting candidate gene given its implication in human VACTERL association as well as in lung development, it is not clear how loss of *DYRK2* function causes a change in *Foxf1* expression or if this is one among many developmentally important genes that are impacted by *Dyrk2* deletion. Rescuing the lung

phenotype with restoration of FOXF1 function would demonstrate the unique importance of this finding.

Reviewer #3 (Remarks to the Author):

In this manuscript, Yogosawa et al. used CRISPR/Cas9 genome editing to generate three distinct *Dyrk2*^{-/-} mouse lines and demonstrate that disruption of *Dyrk2* gene in mice resembles a severe developmental human disorder VATER/VACTERL. They found that *Dyrk2*^{-/-} mice exhibit in reduction of several genes that are linked to VATER/VACTERL in humans. Among these genes was the FOXF1 transcription factor, which regulates epithelial-mesenchymal signaling in the developing lung and other organs. The authors propose that *DYRK2* regulates lung development via FOXF1, and that FOXF1 deficiency is responsible for lung hypoplasia in *Dyrk2*^{-/-} mice. Overall, the experiments are well performed and all three *Dyrk2*^{-/-} mouse lines characterized very rigorously. There is a convincing evidence that *Dyrk2*^{-/-} mice have all phenotypes typical for VATER/VACTERL, including morphological, histological and gene expression changes. The manuscript is well written and easy to follow. *Dyrk2*^{-/-} mice will be useful to study pathogenesis of VATER/VACTERL. Several changes are suggested to improve quality of the manuscript:

Major comments:

1. In Fig. 2G, it is unclear whether cartilage rings are fragmented in *Dyrk2*^{-/-} mice. This can be an artefact of the sectioning. The cartilage rings should be stained, and the whole trachea should be shown from both dorsal and ventral sides to examine cartilage rings.
2. The authors should remove statements from the Results and Discussion sections regarding a possibility of “unrecognized phenotypes in VATER/VACTERL patients”. These statements are speculative even for the Discussion section. Human defects in VATER/VACTERL are extensively characterized using imaging studies, autopsies and histological specimens.
3. Mechanisms of respiratory insufficiency in *Dyrk2*^{-/-} mice remain unclear since the lung has fairly normal histological structure (except a single cyst) (Figs 2J and 5C). FOXF1 deficiency is linked to the loss of alveolar capillaries (alveolar capillary dysplasia) in mice and humans (see a recent reference: PMID 31199666). The authors

should stain Dyrk2^{-/-} lungs for endothelial markers, such as PECAM-1, to determine if the alveolar capillaries are missing or misplaced in the alveolar region. This will explain the respiratory insufficiency in Dyrk2^{-/-} mice.

4. It will be also helpful to stain Dyrk2^{-/-} lungs for endothelial progenitor cells (cKIT⁺PECAM1⁺CD45⁻) that are precursors of capillary endothelial cells. These cells are dependent on FOXF1 (ref, PMID: 31233341) and their loss can explain respiratory insufficiency. FACS is an alternative option to determine the number of cKIT⁺ endothelial progenitor cells.

5. There are no data to show the lung “immaturity” in Dyrk2^{-/-} mice (line 155). This will require the immunostaining for SP-C (type 2 marker) and T1a or Aquaporin 5 (type 1 markers) as well as qRT-PCR for surfactant-associated proteins.

6. It will be helpful to add proliferation/apoptosis studies to identify possible mechanisms of lung hypoplasia in Dyrk2^{-/-} mice.

7. Mechanistically, it is unclear whether DYRK2 can regulate FOXF1. Are these genes expressed in the same cells during embryogenesis? Does DYRK2 have a gradient of expression in the lung mesenchyme, which is similar to the FOXF1 gradient?

Minor comments:

1. Main Figure 1 can be moved to Suppl. Materials.
2. Line 154: figure 5 is presented out of order and between figures 2 and 3.
3. It will be helpful to provide DYRK2 expression pattern during embryonic development.

Point-by-point response to the reviewers:

Thanks for your comments. We appreciate your effort and time. We have carefully considered each point of criticism and suggestion to provide a point-by-point response below. In this revision, the text in blue represents our response to your comments.

Reviewers' comments:

Reviewer #1 (Remarks to the Author):

1. This study generated Dyrk2 null mice via CRISPR and analyzed phenotypes in multiple organs with a particular focus on the lung and claimed a phenotypic similarity to a human disease VATER/VACTERL association. The mutant phenotypes are carefully documented, for 3 versions of the mutation, and provide new information on the role of Dyrk2. However, it is unclear how the phenotypes are connected to the biochemical activity of Dyrk2 and it is unconvincing how similar the phenotypes are to the human disease.

[Response]

Thank you for the suggestion. To connect the phenotypes to the biochemical activity of Dyrk2, we have performed *ex vivo* embryonic lung culture with DYRK kinase inhibitor, harmine. As shown in Fig, 4I, the expression of Foxf1 and its target genes were significantly reduced by treatment with harmine (line 295-297). These data suggest that the kinase activity of Dyrk2 is necessary for the expression of Foxf1 and its target genes in early lung development.

Fig, 4I

2. As pointed out in the second to the last paragraph in Discussion, the mutant mice have phenotypes not observed in the human disease. Additional evidence is needed to support the connection. Is *Dyrk2* mutated or reduced in patients? With the current data, the disease connection is a speculation for Discussion, instead of the main conclusion.

[Response]

This is a very important point. To clarify the connections between the phenotypes of the *Dyrk2*-deficient mice and the features of VATER/VACTERL association patients, we have revised our manuscript and added new Supplementary Data 2 and 3, which summarized several candidate genes and microdeletions in VATER/VACTERL association and mutant mice bearing VATER/VACTERL-like phenotypes. Given the phenotypic diversity of the patients with VATER/VACTERL association, we concluded that the phenotypes of the *Dyrk2*-deficient mice largely corresponds to clinical features of patients with microdeletion/mutation of the *FOXF1* gene.

As the reviewer suggested, we have no information on the *DYRK2* mutated or reduced in patients with this disease. Further studies are thus required to better understand the relationship between the *DYRK2* gene and this rare pediatric disease. We revised our manuscript to moderate the description regarding the disease connection (line 159-164).

3. Although expression microarrays were done on whole embryos, they narrowly focused on VATER/VACTERL-related genes, a few selected genes of lung development, and transcription factors. It's unclear if those genes were selected to make the disease connection. What are the other top differentially expressed genes? Also, how many biological replicates are there for microarray?

[Response]

We have performed microarray analysis once and validated the expression level for the genes of interest by quantitative PCR analysis. The other top differentially expressed genes are described in Fig. S8 (line 194-196).

Fig, S8

4. Which organs and cell types express Dyrk2?

[Response]

Thank you for arising this important question. In this revision, we detected the expression of Dyrk2 in the esophagus, intestine, and kidney as well as the brain, lung, and heart at E14.5 and 18.5 using Western blotting (Figure S1d, e) (line 111-112). We also stained the E18.5 lungs with anti-DYRK2 antibody and detected Dyrk2 signals in epithelial cells of the E18.5 lungs (Figure 4h) (line 285-287). These findings suggest that Dyrk2 may express epithelial cells in early lung development.–

Fig. S1d

Fig. 4h

Fig. S1e

Minor:

5. Fig. 4d: why does the bottom row have fewer genes than the top row?

[Response]

As suggested, we have performed quantitative PCR for the causal genes of human VATER/VACTERL association, *Pten*, *Hoxd13*, *Zic3*, *Fgf8*, *Foxf1*, *Lpp*, *Trap1*, *Pcsk5*, and *Dll3* in both E10.5 (top row) and E14.5 lung (bottom row). However, only 5 genes, *Pten*, *Foxf1*, *Lpp*, *Trap1*, and *Dll3* were expressed in the E14.5 lung. Therefore, the bottom row shows only 5 genes. We have described this information in the main text (line 219-220).

6. Fig. S5d: the mutant pancreas looks different.

[Response]

Thank you for the valuable suggestion. We revised our HE staining for the pancreas and re-evaluated the mutant pancreas, which are now shown in new Fig. S6d and S7d. We, however, detected no significant difference in HE staining of pancreas between the WT and *Dyrk2*^{-/-} embryos.

Fig, S6d

Fig, S7d

7. Table S4: Mouse gene names only capitalize the first letter.

[Response]

As suggested, we corrected mouse gene names only capitalize the first letter.

8. Line 110: other DYRK families should be “family members”?

[Response]

As suggested, we revised this sentence to “family members” in line 110.

9. Line 132: “To prove this hypothesis”. In biomedical research, one does not “prove” but “test”.

[Response]

As indicated, we changed the “prove” to “test” in line 132.

10. Fig. 2b: Please clarify “radial anomalies”.

[Response]

We indicated with arrows in Fig 1b, Fig.S4b and clarified the radial anomalies phenotype.

11. Fig. 5 caption: “through” implies causality, which is not supported by the current data.

[Response]

We agree with the reviewer and revised this sentence to “and the loss of the Foxf1 expression gradient at early lung development”.

Reviewer #2 (Remarks to the Author):

1. Although the authors clearly demonstrate that DYRK2 is required for embryonic development of multiple organs, it is unclear where (in what organs and in which cell types) and at what time points during development DYRK2 is acting. Before presenting the Dyrk2 mutant mouse data it would be helpful to show where Dyrk2 is expressed either by in situ or immunofluorescence staining.

[Response]

Thank you for the comment. As suggested, we have eagerly and thoroughly tried to stain with Dyrk2 in the E11.5 whole embryo in various conditions. However, under all conditions we have examined, no specific signals have been detected in any tissues of the E11.5 whole embryos. Finally, we conducted to immunostain with anti-DYRK2 antibody for E18.5 lungs.

Although at E11.5 lungs, the expression of Dyrk2 was detected only using Western blotting (Figure 4l), we could show the expression of Dyrk2 signals in epithelial cells of the E18.5 lungs (Figure 4h) (line 285-287). We also detected the expression of Dyrk2 in the esophagus, intestine, and kidney as well as the brain, lung, and heart at E14.5 and 18.5 using Western blotting (Figure S1d, e) (line 111-112).

Fig, 4l

Fig. S1d, e

Fig, 4h

2. The link between human VACTERL association and *Dyrk2* loss of function is a central point of this manuscript; however, this link is not clear. Microarray analysis and exome sequencing have both been conducted in patients with VACTERL; however, neither copy number variants including the *DYRK2* locus nor likely pathogenic variants in the *DYRK2* gene have been identified. Furthermore, the phenotype that the *Dyrk2* mutant mice present with includes several defects found in patients with VACTERL association; however, the defects are more severe and include additional anomalies not commonly found in patients with VACTERL. These data suggest that *DYRK2* plays an important role in multiple aspects of development but not necessarily in the mechanisms responsible for VACTERL disease in humans. Decreasing the emphasis on the link with VACTERL and putting more of a focus on the organs affected by *Dyrk2* deletion, and especially the mechanisms responsible for these defects, would be more appropriate.

[Response]

We agree with the reviewer and as suggested, we revised the text to moderate our description regarding the link between human VACTERL association and *Dyrk2* loss of function. We further added description that the loss of *Dyrk2* exhibits developmental abnormalities by regulating the expression pattern of *Foxf1*.

Importantly however, we have clarified the connections between the phenotypes of *Dyrk2*-deficient mice and the features of VATER/VACTERL association patients in new Supplementary Data 2 and 3 that summarize several candidate genes and microdeletions in VATER/VACTERL association and mutant mice featuring VATER/VACTERL-like phenotypes (line 159-164). Given the phenotypic diversity of the patients with VATER/VACTERL association, we concluded that the phenotypes of *Dyrk2*-deficient mice largely correspond to clinical features of patients with microdeletion/mutation of the

FOXF1 gene.

As suggested, neither copy number variants including the *DYRK2* locus nor likely pathogenic variants in the *DYRK2* gene have been identified in the patients. Thus, most likely, the reduction of *Foxf1* could be the main cause of the VATER/VACTERL-L association phenotypes in *Dyrk2*-deficient mice. Further studies are required to better understand the relationship between the *DYRK2* gene and this rare pediatric disease. On the other hand, we have shown that *Dyrk2*-deficient mice exhibit unique phenotypes that are not yet found in patients with VATER/VACTERL-L association, implying potential targets other than *Foxf1*. We added these points in the Discussion (line 177-178, 393).

3. Craniofacial malformations, including cleft palate, are not common in patients with VACTERL; however, 4-5% of patients with esophageal atresia/tracheoesophageal fistula are known to have cleft palate. Some clarifying discussion about this association is warranted. It is also not clear if the *Dyrk2* deletion mice have tracheoesophageal fistula and esophageal atresia (TEF/EA). There appear to be defects in tracheal development and in the esophagus but is there TEF/EA?

[Response]

Thank you for the valuable comment. As shown in Fig. 1g, table 1, and S4g, Supplementary data 1, *Dyrk2* deletion mice have esophageal and tracheal stenosis but not TEF/EA (line 144-146).

Fig, 1g

Fig, S4g

4. The finding of cleft palate in *Dyrk2* mutant mice also makes it unclear what the cause

of neonatal lethality is in these animals. The lungs of the *Dyrk2* mutant mice appear underdeveloped and have cystic malformation of the left lung lobe; however, these defects should not impair their ability to inflate if the pups try to breathe after birth and if the airway is not obstructed. Mice with craniofacial defects, in particular with cleft palate, often die after birth due to malpositioning of the tongue and occlusion of the airway. This results in the phenotype observed the *Dyrk2* mutant mice where the newborn pups attempt to breathe but the lungs fail to inflate as demonstrated by the “sinking lung phenotype”. Either lung/airway specific deletion of *Dyrk2* or bypassing the craniofacial defects via tracheostomy would help to determine if the airway/lung defects or the craniofacial malformation is responsible for neonatal lethality.

[Response]

We agree with the reviewer. To explore the possibility if the airway/lung defects or the craniofacial malformation are responsible for neonatal lethality. We examined the airway/lung defects including lung hypoplasia, immaturity, and tracheal stenosis in addition to cleft palate, which is responsible for neonatal lethality. Indeed, the trachea stenotic phenotype was observed in the *Dyrk2* deletion mice (Fig. 1g, S4g). Therefore, we revised the main text to that respiratory abnormalities including the upper respiratory tract contribute to respiratory failure and neonatal lethality in *Dyrk2*^{-/-} mice (line 262-265).

Fig, 1g

Fig, S4g

5. The phenotype of the *Dyrk2* mice is very interesting but warrants more careful analysis. One general concern is how penetrant the individual defects are? Do all of the mutant mice have the same pattern of defects or is there some variability in the anomalies in

different animals. Also, is there any difference between the 3 different *Dyrk2* mutant lines? The authors show that there is complete loss of DYRK2 protein in all 3 lines. The skeletal defects are not well visualized with micro-CT and might be better demonstrated using skeleton preps with Alcian blue and Alizarin red staining where the entire skeleton can be more easily visualized and measured. This would help to better characterize and demonstrate the craniofacial, vertebral, and limb defects. Alcian blue staining of the trachea would help to better characterize and demonstrate defects in tracheal cartilage development. Similarly, the defects in kidney development are also not clearly demonstrated. What are the structures in the *Dyrk2* mutant kidneys that are not normally developed?

[Response]

Thank you for the insightful suggestions. We have included the summary of the abnormal phenotypes in 3 different *Dyrk2*^{-/-} mice lines in Supplementary data 1 (line 156-159). There is no significant difference among 3 different *Dyrk2* mutant lines. We added the data of skeletal preparations by Alcian Blue/Alizarin Red staining instead of micro-CT (Fig 1b, c, 2a, S4b, c, S5a). We have also demonstrated the defects in kidney development (Fig 1h, S4h) (line 148).

Fig. 1b, c

Fig. 2a

Fig. S4b, c

Fig. S5a

Fig. 1h

Fig. S4h

6. Although the authors do not investigate or speculate on the cellular mechanisms responsible for the multiple anomalies in the *Dyrk2* mutant mice, their background studying *Dyrk2* regulation of cell cycling in cancer biology would suggest a hypothesis where loss of *Dyrk2* function during development might cause increased or inappropriate cell proliferation and loss of cellular differentiation. Instead the authors report that *Dyrk2* mutant mice show hypoplasia of multiple structures. Could the authors demonstrate or speculate on the mechanisms responsible for the generalized hypoplasia found in multiple organs in the *Dyrk2* mutant mice and how this differs from the role of *Dyrk2* in regulation of cell cycling and tumor formation?

[Response]

As shown in our recent published paper (Yoshida et al., *Elife*. 9:e57381. (2020)), the loss of *Dyrk2* in mice causes suppression of Sonic hedgehog (Shh) signaling. Since *Foxf1* is a target of Shh signaling, *Dyrk2* is required to form a subepithelial-to-distal expression gradient of *Foxf1* through the suppression of Shh signaling. In concert with this idea, the *Dyrk2*-deficient mice exhibit decreased lung proliferation (Fig. 4 m) (line 298-299). This finding indicates that the substrates of DYRK2 may be different in tumor formation and embryogenesis.

In future studies, any findings of novel substrates for *Dyrk2* during embryogenesis would contribute to understanding detailed mechanisms of lung hypoplasia in the *Dyrk2* mutant mice.

Fig, 4m

7. To understand the genetic mechanisms responsible for the malformations found in *Dyrk2* mutant mice, the authors conducted gene expression array analysis and qRT-PCR experiments to identify changes in gene expression associated with these developmental anomalies. The selection of the developmental timepoints at which these gene expression experiments were conducted was based on the hypothesis that differences would be present during “early organogenesis” at embryonic day 8.5 and 10.5; however, it is unclear if any morphological defects were evident in *Dyrk2* mutant mice at these time points. Dissection and whole mount or histological analysis of *Dyrk2* mutant embryos would help to determine when defects in development are first evident. These experiments would also help to better determine at what developmental timepoint(s) experiments to identify changes in gene expression should be conducted to identify expression changes that might be responsible for the observed developmental defects.

[Response]

Thank you for a helpful suggestion. To determine when the first defect appears in development, we examined the primordial endodermal epithelium at E10.5 with whole-mount E-cadherin immunostaining. As shown in Fig. 3f, there is no obvious morphological defect in the primordial trachea, esophagus, lung, stomach and intestine of *Dyrk2*^{-/-} embryos at this timepoint. Because the *Dyrk2*^{-/-} embryos initially exhibit genetic abnormalities around E8.5, we would propose that the genetic differences appear during “early organogenesis” embryonic days from 8.5 to 10.5, which causes the morphological defects of endodermal organs after E11.5 (Fig. 4f) (line 234-237).

Fig. 3f

Fig. 4f

8. Given the spectrum malformations identified in the *Dyrk2* mutant mice, it is not surprising that the authors found multiple gene expression changes in transcription factor genes. The authors focused on changes in gene expression where the genes of interest were decreased in expression; however, increases in gene expression are also likely to cause abnormal development. Given the authors background with *Dyrk2*, any evidence or speculation about how loss of *DYRK2* function might directly affect gene expression would help to better understand the genetic mechanisms responsible for the genetic and developmental abnormalities observed in the *Dyrk2* mutant mice. Furthermore, a more complete analysis of the identified gene expression changes would be helpful as well as eventual depositing of the gene expression array results in a transcriptome analysis database such as the gene expression omnibus (GEO).

[Response]

Thank you for the valuable suggestion. We conducted quantitative PCR for the top differentially increased genes and added the results in Fig. S8c (line 194-196).

Although we observed increased levels in gene expression, such as *Gata1*, *Lyl1*, and *Ikzf1*, which related to lymphocyte and erythrocyte development, functional analyses for these genes are beyond the focus in this paper. Further analyses would be conducted in the future study.

We believe that the cause of the major phenotypes in the *Dyrk2*-deficient mice are the reduction of Foxf1 expression. We show that the expression of Foxf1 and its target genes was significantly reduced by treatment with harmine, DYRK kinase inhibitor, in *ex vivo* embryonic lung culture (Figure 4l). These data suggest that the kinase activity of Dyrk2 is necessary for the expression of Foxf1 and its target genes in early lung development.

We have recently reported that loss of *Dyrk2* in mice causes suppression of Shh signaling (Yoshida et al., *Elife*. 9:e57381. (2020)). Since Foxf1 is a target of Shh signaling, *Dyrk2* could be required to form a subepithelial-to-distal expression gradient of Foxf1 by regulating epithelial-to-mesenchymal Shh signaling.

we have already deposited the gene expression array to GEO as shown in the Method section (accession no. GSE146614) (line 487).

Fig. S8c

9. The authors focus on the developmental defects associated with lung hypoplasia due to their hypothesis that lung hypoplasia is responsible for neonatal lethality in *Dyrk2* mutant mice. Deletion of *Dyrk2* appears to impact both branching morphogenesis and

septation of the gas exchange regions of the lung. It would be helpful for the authors to show where (in which cell populations) and when during development *Dyrk2* expression is evident in the airway and lungs. Furthermore, because of the focus on lung hypoplasia, a more thorough demonstration of when these defects in lung development become evident and whether *Dyrk2* is affecting cell cycling is important and worthwhile.

[Response]

We agree with these comments. We conducted to immunostain with anti-DYRK2 antibody for E11.5 lungs.

Although at E11.5 lungs, the expression of *Dyrk2* was detected only using Western blotting (Figure 4l), we have shown the expression of *Dyrk2* signals in epithelial cells of the E18.5 lungs (Figure 4h). These findings suggest that *Dyrk2* may express epithelial cells in early lung development.

As shown in the response to Reviewer #2 comment No. 7, the foregut epithelium of embryos displayed no significant differences between the WT and *Dyrk2*^{-/-} embryos at E10.5 (Figure 3f). This finding indicates that the *Dyrk2*^{-/-} embryos initially exhibit genetic abnormalities around E8.5-10.5 that may be responsible for the developmental defects. Therefore, we demonstrated that the defects of *Dyrk2*^{-/-} lungs initially occurred from E11.5.

We have stained with Ki67 and cleaved caspase 3 for proliferation/apoptosis studies. As shown in Fig, 4m, the proliferation and apoptosis rate decreased in the E11.5 *Dyrk2*^{-/-} lung, suggesting that the *Dyrk2*^{-/-} embryos exhibited lung hypoplasia from E11.5 by decreasing cell proliferation (line 298-299).

Fig, 4m

10. Finally, the authors conclude that deletion of *Dyrk2* has a direct or possibly secondary impact on *Foxf1* gene expression and regulation of a subepithelial protein gradient. Although *Foxf1* is clearly an interesting candidate gene given its implication in human VACTERL association as well as in lung development, it is not clear how loss of *DYRK2* function causes a change in *Foxf1* expression or if this is one among many developmentally important genes that are impacted by *Dyrk2* deletion. Rescuing the lung phenotype with restoration of *FOXF1* function would demonstrate the unique importance of this finding.

[Response]

We agree with the reviewer. In this paper, we failed to determine molecular mechanism by which loss of *Dyrk2* function changes in *Foxf1* expression. In this revision, however, we show that the kinase activity of *Dyrk2* is necessary for the *Foxf1* expression. Additionally, we have recently shown that loss of *Dyrk2* in mice causes suppression of *Shh* signaling (Yoshida et al., *Elife*. 9:e57381. (2020)). Since *Foxf1* is a target of *Shh* signaling, we speculate that *Dyrk2* affects *Shh* signaling in lung epithelial cells, which leads to reduce the subepithelial-to-distal expression gradient of *Foxf1* in the lung mesenchyme. We have included these discussions in the main text (line 381-386).

To clarify whether the *Foxf1* is one of developmentally important genes that are impacted by *Dyrk2* deletion, we would need to generate a new transgenic mouse line expressing *Foxf1* in developing lung mesoderm to rescue the phenotypes of *Dyrk2*-deficient mice. Novel substrates of *Dyrk2* in lung epithelial cells during lung development also need to find out. These studies would be carried out in the separate study.

Reviewer #3 (Remarks to the Author):

Major comments:

1. In Fig. 2G, it is unclear whether cartilage rings are fragmented in *Dyrk2*^{-/-} mice. This can be an artefact of the sectioning. The cartilage rings should be stained, and the whole trachea should be shown from both dorsal and ventral sides to examine cartilage rings.

[Response]

Thank you for the suggestion. We conducted Alcian blue staining of whole trachea to see

the entire structure of the cartilages. The fragmented cartilage phenotype was indeed observed at both the dorsal and ventral sides. We added these data in Fig. 1g and S4g (line 144-146).

Fig. 1g

Fig. S4g

2. The authors should remove statements from the Results and Discussion sections regarding a possibility of “unrecognized phenotypes in VATER/VACTERL patients”. These statements are speculative even for the Discussion section. Human defects in VATER/VACTERL are extensively characterized using imaging studies, autopsies and histological specimens.

[Response]

We agree with the reviewer and as indicated, we removed the description regarding a possibility of “unrecognized phenotypes in VATER/VACTERL patients” from the main text and revised to “our *Dyrk2*^{-/-} mice provide a better understanding of embryogenesis” (line 177-178, 393, 397-398).

3. Mechanisms of respiratory insufficiency in *Dyrk2*^{-/-} mice remain unclear since the lung has fairly normal histological structure (except a single cyst) (Figs 2J and 5C). FOXF1 deficiency is linked to the loss of alveolar capillaries (alveolar capillary dysplasia) in mice and humans (see a recent reference: PMID 31199666). The authors should stain *Dyrk2*^{-/-} lungs for endothelial markers, such as PECAM-1, to determine if the alveolar capillaries are missing or misplaced in the alveolar region. This will explain the respiratory insufficiency in *Dyrk2*^{-/-} mice.

[Response]

This is a very important question to uncover the cause of respiratory insufficiency in *Dyrk2*^{-/-} mice. We have stained the E18.5 *Dyrk2*^{-/-} lung for the endothelial markers VEGFR2. However, there was no significant difference in VEGFR2 staining between the WT and *Dyrk2*^{-/-} lung (Fig. 4d). On the other hand, as we responded to the Reviewer #3 comment No. 5, the expression of Podoplanin, the alveolar type 1 cell marker, were decreased in the E18.5 *Dyrk2*^{-/-} lung, suggesting that the *Dyrk2*^{-/-} embryos exhibited lung immaturity (Fig. 4d). Furthermore, as we responded to the Reviewer #2 comment No. 4, *Dyrk2*^{-/-} mice have the tracheal stenosis phenotype (Fig. 1g). Thus, the cause of respiratory insufficiency in *Dyrk2*^{-/-} mice might be related to lung immaturity and trachea stenosis. We added these results in the main text (line 259-265).

Fig. 4d

4. It will be also helpful to stain *Dyrk2*^{-/-} lungs for endothelial progenitor cells (cKIT⁺PECAM1⁺CD45⁻) that are precursors of capillary endothelial cells. These cells are dependent on FOXF1 (ref, PMID: 31233341) and their loss can explain respiratory insufficiency. FACS is an alternative option to determine the number of cKIT⁺ endothelial progenitor cells.

[Response]

As suggested, we have stained the E18.5 *Dyrk2*^{-/-} lung for endothelial markers, VEGFR2, as shown in the Reviewer #3 comment No. 3. However, there was no significant difference in VEGFR2 staining between the WT and *Dyrk2*^{-/-} lung. Therefore, we concluded that the *Dyrk2*^{-/-} embryos exhibited no abnormalities in endothelial cells and those progenitor cells.

5. There are no data to show the lung “immaturity” in *Dyrk2*^{-/-} mice (line 155). This will require the immunostaining for SP-C (type 2 marker) and T1a or Aquaporin 5 (type 1 markers) as well as qRT-PCR for surfactant-associated proteins.

[Response]

As indicated, we performed the immunostaining for Podoplanin (type 1 marker) and Pro-SP-C (type 2 marker) as well as qRT-PCR for type 1 and 2 markers. Podoplanin expression were decreased in the E18.5 *Dyrk2*^{-/-} lung, suggesting that the *Dyrk2*^{-/-} embryos exhibited lung immaturity. We added these data in Fig. 4d and e (line 259-262).

Fig. 4d, e

6. It will be helpful to add proliferation/apoptosis studies to identify possible mechanisms of lung hypoplasia in *Dyrk2*^{-/-} mice.

[Response]

Thank you for the helpful comment. We examined Ki67 and cleaved caspase 3 in E11.5 to determine proliferation/apoptosis in both WT and mutant. As shown in Fig. 4m, both proliferation and apoptosis rate were decreased in the *Dyrk2*^{-/-} lung, implying that the *Dyrk2*^{-/-} embryos exhibited lung hypoplasia through decreased cell proliferation (line 297-299).

Fig, 4m

7. Mechanistically, it is unclear whether DYRK2 can regulate FOXF1. Are these genes expressed in the same cells during embryogenesis? Does DYRK2 have a gradient of expression in the lung mesenchyme, which is similar to the FOXF1 gradient?

[Response]

This is an important question. As we responded to reviewer #2, we stained the E18.5 lungs with anti-DYRK2 antibody and identified airway epithelial cells expressing *Dyrk2* (Figure 4h) (line 285-287). Because *Foxf1* is expressed in mesenchymal cells, *Dyrk2* expressing cells are distinct from *Foxf1* expressing cells, suggesting indirect regulation of the mesenchymal *Foxf1* gradient by epithelial *Dyrk2*.

The expression of *Foxf1* and its target genes was significantly reduced by DYRK inhibitor treatment (Fig. 4l). These data suggest that the kinase activity of *Dyrk2* is necessary for the indirect regulation of *Foxf1*. Furthermore, in our recent publication, we found that loss of *Dyrk2* in mice causes suppression of *Shh* signaling (Yoshida et al., *Elife*. 9:e57381. (2020).

In the present study, we confirmed that *Dyrk2* modulates *Shh* expression level in lung epithelial cells (Fig. 3d). Loss of *Dyrk2* led to reduction of *Shh* and altered *Foxf1* expression pattern in the lung mesenchyme. *Dyrk2* could be required to form a subepithelial-to-distal expression gradient of *Foxf1* by regulating epithelial-to-mesenchymal *Shh* signaling. Further studies are required to understand the mechanism of the *Dyrk2*-mediated *Shh*-*Foxf1* signaling regulation.

We thus revised the main text and added these points in the Discussion (line 381-386).

Fig, 4l

Fig, 4h

Minor comments:

1. Main Figure 1 can be moved to Suppl. Materials.

[Response]

As indicated, we have moved Fig. 1 to Fig. S1.

2. Line 154: figure 5 is presented out of order and between figures 2 and 3.

[Response]

As indicated, we have removed “figure 5” (Line 154).

3. It will be helpful to provide DYRK2 expression pattern during embryonic development.

[Response]

As indicated, we have stained with Dyrk2. Dyrk2 was expressed in epithelial cells in the E18.5 lungs (line 286-287). We also detected the expression of Dyrk2 in the esophagus, intestine, and kidney as well as the brain, lung, and heart at E14.5 and 18.5 using Western blotting (Figure S1d, e) (line 111-112).

Fig, 4h

Fig. S1d, e

Reviewers' comments:

Reviewer #1 (Remarks to the Author):

I am satisfied with the revision.

Reviewer #2 (Remarks to the Author):

Thank you for the chance to review this revised manuscript. The authors have done an excellent job of increasing their phenotypic analysis of the Dyrk2 mutant embryonic mice. Despite the significant improvement in the manuscript there several issues that preclude me from recommending that this manuscript be published.

Major concerns:

1. The central argument that the authors make is that the genetic and developmental mechanisms of complex congenital diseases, such as VATER association, remain poorly understood. That is true. The authors propose that Dyrk2 mutant mice represent a novel animal model of VATER (Line 166). This conclusion is misleading and a major distraction from the many interesting and worthwhile aspects of the manuscript. The authors nicely show that Dyrk2 mutants have many interesting defects in development including defects in tissues that are commonly abnormal in patients with VATER. The defects in these tissues in the Dyrk2 mutant mice are not the typical defects observed in patients with VATER and tend to be more severe. Additionally, the Dyrk2 mutant mice have many other malformations that are not included in VATER association. A more accurate conclusion based on the mutant mouse phenotype analysis is that Dyrk2 is essential for embryonic development of multiple tissues and that loss of Dyrk2 results in lethality after birth.

The authors' suggestion that Dyrk2 mutant mice represent a novel animal model of human VATER association is misleading and would be confusing to clinicians, basic researchers, and physician-scientists who are trying to better understand the links between human genetic variants and the molecular mechanisms responsible for disease. Mentioning VATER in the context of the evaluation of Dyrk2 potentially mis-regulating FoxF1 might be appropriate; however, the focus on VATER throughout the manuscript and in the title is not warranted and inaccurate.

2. The most interesting aspect of the manuscript is the role of Dyrk2 during development, especially in the development of the lungs and airway as the Dyrk2 mutant mice appear to die after birth with respiratory failure. This makes the major strength of the paper its phenotype analysis. To begin to understand the mechanisms responsible for the phenotype, the authors need to show where Dyrk2 is expressed. The authors state that they attempted to show the pattern of Dyrk2 expression using multiple approaches but only show western blot and limited IF staining. An alternative approach would be to use RNA-scope or insitu hybridization. Without this limited understanding of which cells express Dyrk2 and at which stages of development, it is not possible to analyze the important cellular or developmental mechanisms responsible for the phenotype.

3. Although the authors have increased their analysis of the abnormal phenotype, there is very limited investigation of the potential cellular or developmental mechanisms that might be responsible for the phenotypes. The authors have done some analysis of the cell types present in the gas-exchange region of the late embryonic lung; however, they also show that there are defects very early in lung and airway development. Focusing on those early time points with more detailed analysis of the cell types present as those general anatomic defects become evident is warranted and would help to better understand the role that Dyrk2 plays.

4. Aside from the analysis and description of the phenotype, the second most important potential

contribution of this manuscript is the role that Dyrk2 might play in regulating other genes known to be important for lung or airway development. The authors have made some modifications to their analysis and discussion of the potential genetic mechanisms responsible for the observed lung phenotypes in the Dyrk2 mutant mice. They focus primarily on a proposed genetic mechanism involving downregulation of Foxf1 because of the association of FOXF1 variants in humans with VATER and the role of Foxf1 in lung development. They also showed that kinase activity is necessary for the expression of Foxf1. These are interesting findings however the suggestion that down-regulation of Foxf1 is the primary mechanism responsible for the lung phenotypes is speculative and not adequately supported by the data. Without a more clear investigation of how Dyrk2 directs Foxf1 expression, an analysis of a cellular or developmental mechanism that involves Foxf1, or a more clear genetic or molecular rescue experiment that shows that restoration of Foxf1 expression or molecular function results in normalization of development (if done in vivo – a more challenging experiment as the authors acknowledge in their rebuttal) or a key cellular mechanism (this could be done in an in vitro cell culture model) there is not sufficient novel data to support a conclusion that the genetic interaction between Dyrk2 and Foxf1 is important in the mechanisms responsible for the Dyrk2 mutant phenotype.

Reviewer #3 (Remarks to the Author):

All my comments were addressed. The manuscript has been improved after revision.

Reviewers' comments:

Reviewer #1 (Remarks to the Author):

I am satisfied with the revision.

Reviewer #2 (Remarks to the Author):

Thank you for the chance to review this revised manuscript. The authors have done an excellent job of increasing their phenotypic analysis of the Dyrk2 mutant embryonic mice. Despite the significant improvement in the manuscript there several issues that preclude me from recommending that this manuscript be published.

Major concerns:

1. The central argument that the authors make is that the genetic and developmental mechanisms of complex congenital diseases, such as VATER association, remain poorly understood. That is true. The authors propose that Dyrk2 mutant mice represent a novel animal model of VATER (Line 166). This conclusion is misleading and a major distraction from the many interesting and worthwhile aspects of the manuscript. The authors nicely show that Dyrk2 mutants have many interesting defects in development including defects in tissues that are commonly abnormal in patients with VATER. The defects in these tissues in the Dyrk2 mutant mice are not the typical defects observed in patients with VATER and tend to be more severe. Additionally, the Dyrk2 mutant mice have many other malformations that are not included in VATER association. A more accurate conclusion based on the mutant mouse phenotype analysis is that Dyrk2 is essential for embryonic development of multiple tissues and that loss of Dyrk2 results in lethality after birth.

The authors' suggestion that Dyrk2 mutant mice represent a novel animal model of human VATER association is misleading and would be confusing to clinicians, basic researchers, and physician-scientists who are trying to better understand the links between human genetic variants and the molecular mechanisms responsible for disease. Mentioning VATER in the context of the evaluation of Dyrk2 potentially mis-regulating FoxF1 might be appropriate; however, the focus on VATER throughout the manuscript and in the title is not warranted and inaccurate.

2. The most interesting aspect of the manuscript is the role of Dyrk2 during development, especially in the development of the lungs and airway as the Dyrk2 mutant mice appear to die after birth with respiratory failure. This makes the major strength of the paper its phenotype analysis. To begin to understand the mechanisms responsible for the phenotype, the authors need to show where Dyrk2 is expressed. The authors state that they attempted to show the pattern of Dyrk2 expression using multiple approaches but only show western blot and limited IF staining. An alternative approach would be to use RNA-scope or insitu hybridization. Without this limited understanding of which cells express Dyrk2 and at which stages of development, it is not possible to analyze the important cellular or developmental mechanisms responsible for the phenotype.

3. Although the authors have increased their analysis of the abnormal phenotype, there is very limited investigation of the potential cellular or developmental mechanisms that might be responsible for the phenotypes. The authors have done some analysis of the cell types present in the gas-exchange region of the late embryonic lung; however, they also show that there are defects very early in lung and airway development. Focusing on those early time points with more detailed analysis of the cell types present as those general anatomic defects become evident is warranted and would help to better understand the role that Dyrk2 plays.

4. Aside from the analysis and description of the phenotype, the second most important potential contribution of this manuscript is the role that Dyrk2 might play in regulating other genes known to be important for lung or airway development. The authors have made some modifications to their analysis and discussion of the potential genetic mechanisms responsible for the observed lung phenotypes in the Dyrk2 mutant mice.

They focus primarily on a proposed genetic mechanism involving downregulation of Foxf1 because of the association of FOXF1 variants in humans with VATER and the role of Foxf1 in lung development. They also showed that kinase activity is necessary for the expression of Foxf1. These are interesting findings however the suggestion that downregulation of Foxf1 is the primary mechanism responsible for the lung phenotypes is speculative and not adequately supported by the data.

Without a more clear investigation of how Dyrk2 directs Foxf1 expression, an analysis of

a cellular or developmental mechanism that involves Foxf1, or a more clear genetic or molecular rescue experiment that shows that restoration of Foxf1 expression or molecular function results in normalization of development (if done in vivo – a more challenging experiment as the authors acknowledge in their rebuttal) or a key cellular mechanism (this could be done in an in vitro cell culture model) there is not sufficient novel data to support a conclusion that the genetic interaction between Dyrk2 and Foxf1 is important in the mechanisms responsible for the Dyrk2 mutant phenotype.

Reviewer #3 (Remarks to the Author):

All my comments were addressed. The manuscript has been improved after revision.

Point-by-point response to the reviewers:

Thanks for your comments. We appreciate your effort and time. We have carefully considered each point of criticism and suggestion to provide a point-by-point response below. In this revision, the text in blue represents our response to your comments.

Reviewers' comments:

Reviewer #1 (Remarks to the Author):

I am satisfied with the revision.

Reviewer #2 (Remarks to the Author):

Thank you for the chance to review this revised manuscript. The authors have done an excellent job of increasing their phenotypic analysis of the Dyrk2 mutant embryonic mice. Despite the significant improvement in the manuscript there several issues that preclude me from recommending that this manuscript be published.

Major concerns:

1. The central argument that the authors make is that the genetic and developmental mechanisms of complex congenital diseases, such as VATER association, remain poorly understood. That is true. The authors propose that Dyrk2 mutant mice represent a novel animal model of VATER (Line 166). This conclusion is misleading and a major distraction from the many interesting and worthwhile aspects of the manuscript. The authors nicely show that Dyrk2 mutants have many interesting defects in development including defects in tissues that are commonly abnormal in patients with VATER. The defects in these tissues in the Dyrk2 mutant mice are not the typical defects observed in patients with VATER and tend to be more severe. Additionally, the Dyrk2 mutant mice have many other malformations that are not included in VATER association. A more accurate conclusion based on the mutant mouse phenotype analysis is that Dyrk2 is essential for embryonic development of multiple tissues and that loss of Dyrk2 results in lethality after birth.

The authors' suggestion that Dyrk2 mutant mice represent a novel animal model of human VATER association is misleading and would be confusing to clinicians, basic researchers, and physician-scientists who are trying to better understand the links between human genetic variants and the molecular mechanisms responsible for disease. Mentioning VATER in the context of the evaluation of Dyrk2 potentially mis-regulating FoxF1 might be appropriate; however, the focus on VATER throughout the manuscript and in the title is not warranted and inaccurate.

[Response]

We agree with the reviewer and as suggested, we corrected the title as well as the text to focus on the significance of Dyrk2 for embryogenesis, especially its importance in lung development. Furthermore, we would like to propose that Dyrk2 acts as a positive regulator of the Shh-Foxf1 interaction to generate the subepithelial-to-distal expression gradient of Foxf1 during lung development.

We modified the title from “DYRK2-deficiency exhibits congenital malformations of multiple organs observed in human VATER/VACTERL-L association” to “DYRK2-deficiency exhibits congenital malformations with lung hypoplasia by impaired Shh-Foxf1 axis”.

2. The most interesting aspect of the manuscript is the role of Dyrk2 during development, especially in the development of the lungs and airway as the Dyrk2 mutant mice appear to die after birth with respiratory failure. This makes the major strength of the paper its phenotype analysis. To begin to understand the mechanisms responsible for the phenotype, the authors need to show where Dyrk2 is expressed. The authors state that they attempted to show the pattern of Dyrk2 expression using multiple approaches but only show western blot and limited IF staining. An alternative approach would be to use RNA-scope or in situ hybridization. Without this limited understanding of which cells express Dyrk2 and at which stages of development, it is not possible to analyze the important cellular or developmental mechanisms responsible for the phenotype.

[Response]

As requested, we re-examined the cell type in which Dyrk2 is expressed during lung

development.

Since Dyrk2-deficient mouse was generated using the CRISPR/Cas9 nickase system, alternative approach such as RNA-scope or in situ hybridization cannot detect the Dyrk2 mRNA signal difference between WT and Dyrk2-deficient mice. Therefore, we performed immunohistochemistry for Dyrk2 with CC10 (club cell marker in E18.5 lung) and FoxJ1/Acetylated tubulin (ciliated cell marker in E16.5, and 18.5 lung).

As shown in new Figure 4c-e, we have been able to detect the expression of Dyrk2 signals in airway epithelium at E11.5 and E18.5 lungs. Moreover, as shown in new Figure 4d and e, Dyrk2 is expressed in ciliated cells in the late embryonic lung (E16.5, 18.5). We revealed that Dyrk2 is expressed in epithelial cells throughout lung development. We added this data as new Figure 4c-e (line 254-256).

New Fig, 4c

New Fig, 4d, e

3. Although the authors have increased their analysis of the abnormal phenotype, there is very limited investigation of the potential cellular or developmental mechanisms that might be responsible for the phenotypes. The authors have done some analysis of the cell types present in the gas-exchange region of the late embryonic lung; however, they also show that there are defects very early in lung and airway development. Focusing on those early time points with more detailed analysis of the cell types present as those general anatomic defects become evident is warranted and would help to better understand the role that Dyrk2 plays.

[Response]

We agree with the reviewer’s suggestion and as shown in the response to comment No. 2, we detected Dyrk2 protein earlier than the cell types in the gas-exchange region of the late embryonic lung.

As shown in new Figure 4c, e, we have been able to confirm that Dyrk2 is expressed in ciliated cells (Acetylated tubulin-positive cells) in the late embryonic E16.5 lung. We have also shown that Dyrk2 is expressed in epithelial cells in E11.5 lung. Taken together, we would like to respond to Reviewer #2’s comment No. 3 as follows:

“Dyrk2 expresses in epithelial cells at early lung development. Given the kinase activity of Dyrk2 is required for Foxf1 expression (Figure 4i) and the loss of Dyrk2 results in the downregulation of Shh expression at early lung development (Figure 3d), Dyrk2 might regulate epithelial-to-mesenchymal interaction via inducing Shh ligand expression dependent on its kinase activity. (line 308-309)” We added this data as new Figure 4c and e (line 254-256).

New Fig, 4c

New Fig, 4e

4. Aside from the analysis and description of the phenotype, the second most important potential contribution of this manuscript is the role that Dyrk2 might play in regulating other genes known to be important for lung or airway development. The authors have made some modifications to their analysis and discussion of the potential genetic mechanisms responsible for the observed lung phenotypes in the Dyrk2 mutant mice.

They focus primarily on a proposed genetic mechanism involving downregulation of Foxf1 because of the association of FOXF1 variants in humans with VATER and the role of Foxf1 in lung development. They also showed that kinase activity is necessary for the expression of Foxf1. These are interesting findings however the suggestion that downregulation of Foxf1 is the primary mechanism responsible for the lung phenotypes is speculative and not adequately supported by the data.

Without a more clear investigation of how Dyrk2 directs Foxf1 expression, an analysis of a cellular or developmental mechanism that involves Foxf1, or a more clear genetic or molecular rescue experiment that shows that restoration of Foxf1 expression or molecular function results in normalization of development (if done in vivo – a more challenging

experiment as the authors acknowledge in their rebuttal) or a key cellular mechanism (this could be done in an in vitro cell culture model) there is not sufficient novel data to support a conclusion that the genetic interaction between Dyrk2 and Foxf1 is important in the mechanisms responsible for the Dyrk2 mutant phenotype.

[Response]

As suggested by Reviewer #2, down-regulation of Foxf1 may not be the primary mechanism responsible for the lung phenotype. However, we show that Dyrk2^{-/-} mice exhibit lung malformations likewise the phenotypes of Foxf1^{+/-} mice.

As described in the main text, Dyrk2^{-/-} mice exhibit lung defects such as hypoplasia (Figure 1m), fusion of right lung lobes (Figure 1m), esophageal and tracheal stenosis (Figure 1k), the hypoplastic tracheal cartilage (Figure 1k), and airway branching defects (Figure 4a, b). We show that Dyrk2 KO results in significant reduction of Foxf1 expression (Figure 4f, g).

Figure 1m

Figure 1k

Figure 4a, b

Figure 4f, g

Similarly, it was already reported that haploinsufficiency of Foxf1 causes lung malformations likewise the phenotypes with Dyrk2-deficient mice (Mahlapuu et al., Development. 2001 Jun;128(12):2397-406., Lim et al., Am J Physiol Lung Cell Mol Physiol. 2002 May;282(5):L1012-22).

Reference: Development. 2001 Jun;128(12):2397-406.

“Haploinsufficiency of the forkhead gene Foxf1, a target for sonic hedgehog signaling, causes lung and foregut malformations”

This paper demonstrated that the haploinsufficiency of Foxf1 causes a variable phenotype that includes lung immaturity and hypoplasia, fusion of right lung lobes, narrowing of esophagus and trachea, the hypoplastic tracheal cartilage, and airway branching defects.

Reference: Am J Physiol Lung Cell Mol Physiol. 2002 May;282(5):L1012-22

“Fusion of lung lobes and vessels in mouse embryos heterozygous for the forkhead box f1 targeted allele”

This paper demonstrated that Foxf1^{-/-} embryos exhibit lung defects such as hypoplasia, fusion of right lung lobes, and airway branching defects.

Furthermore, to determine when the first defect appears in development, we examined the primordial endodermal epithelium at E10.5 with whole-mount E-cadherin immunostaining. As shown in Fig. 2e, there was no obvious morphological defect in the primordial trachea, esophagus, lung, stomach and intestine of *Dyrk2*^{-/-} embryos at this timepoint. Finally, we would argue that the genetic differences appear during “early organogenesis” embryonic days from 8.5 to 10.5, which causes the morphological defects of endodermal organs after E11.5 (Fig. 4a). Therefore, we would like to propose that down-regulation of *Foxf1* is the main cause for the *Dyrk2*^{-/-} early lung phenotypes.

Figure 2e

Figure 4a

Whereas it is still possible that other targets (for example, lung development-related genes such as *Shh*, *Fgf10*, *Foxa2*, *Notch1*, *Foxp2*, and *Nkx2.1* as shown in Figure 2c, d) of *Dyrk2* would be involved in the morphogenesis defects in the mutant, we would like to propose that altered *Dyrk2*-*Foxf1* axis is a promising pathway as a cause of the lung hypoplasia phenotype. We added these points in the Discussion (line 311-312).

Furthermore, as requested, we investigated how *Dyrk2* regulates *Foxf1* expression. Previous reports have shown that in mouse models, *Foxf1* transcription in the lung mesenchyme is activated by epithelial *Shh* and is required for airway branching morphogenesis (*Development*. 2001 Jun;128(12):2397-406., *Am J Physiol Lung Cell Mol Physiol*. 2002 May;282(5):L1012-22). Genetic studies with mice have demonstrated that *Foxf1* acts downstream of *Shh*-Gli signaling via epithelial-to-mesenchymal interaction during organogenesis (*Development*. 2001 Jun;128(12):2397-406., *Development* 2001 128, 155-166., *J Biol Chem* 2005 280, 37908-37916.)

Indeed in our manuscript, we demonstrated that *Shh* expression was reduced in E10.5 *Dyrk2*^{-/-} embryos (Figure 2d). The kinase activity of *Dyrk2* was further necessary for the indirect regulation of *Foxf1* in E11.5 lung (Figure 4i). Based on these findings, we would

like to propose that an interaction between Dyrk2 and Shh-Foxf1 signaling is particularly important in lung development. Dyrk2 might regulate the epithelial-to-mesenchymal interaction via activating Shh ligand expression dependent on its kinase activity.

Figure 2d

Figure 4i

Therefore, we planned to assess whether the activation Shh signaling in (1) the Dyrk2^{-/-} cells (mouse embryonic fibroblast (MEF)) or (2) the Dyrk2^{-/-} lungs, restores Foxf1 expression and the Foxf1 target genes in an in vivo or in vitro cell culture model.

(1) We first investigated if this question could be solved by using MEF.

As shown below, in an in vitro MEF culture model, neither WT nor Dyrk2-deficient cells can be examined because Foxf1 expression was not detected (Figure R1). We thus concluded that it is not appropriate to examine the interaction between Dyrk2 and Shh-Foxf1 signaling in this model.

Figure R1. Expression levels of Dyrk2 and Foxf1 in E11.5 lung explants or MEF with WT and Dyrk2^{-/-} by western blotting

(2) We thus figured out whether activation of Shh signal in Dyrk2 mutant lungs restores Foxf1 expression and the Foxf1 target genes in an ex vivo E11.5 lung culture model.

As shown in new Figure 4j, Shh activation by Smoothened agonist (SAG), a potent pharmacological activator of the shh co-receptor Smoothened (Dev Cell. 2015 Nov 9;35(3):322-32.), in the Dyrk2 mutant lungs partially but robustly restored Foxf1 expression and the Foxf1 target genes. This finding supports the conclusion that an interaction between Dyrk2 and Shh-Foxf1 signaling is critical for the lung development. We added this result in the main text (line 266-271).

New Figure 4j

Reviewer #3 (Remarks to the Author):

All my comments were addressed. The manuscript has been improved after revision.

REVIEWERS' COMMENTS:

Reviewer #2 (Remarks to the Author):

Thank you very much again for the chance to review the manuscript "Dyrk2-deficiency exhibits congenital malformations with lung hypoplasia by impaired Shh-Foxf1 axis". The authors have done a very good job of addressing the prior review and their manuscript is significantly improved. The authors have demonstrated that Dyrk2 plays an important role in the development of multiple organs in mice and that, in the lung, Dyrk2 plays an important role in regulating an essential Shh/Foxf1 molecular pathway. My suggestions are minor and can be addressed by word editing only.

The manuscript is well written and generally the text is clear; however, there are many minor tense or singular/plural mistakes that can be addressed by editing.

In the introduction there is a typo on line 69 ("format" instead of "form at").

In the results on line 103 the authors might consider changing the text from "little is known about the impact of Dyrk2 gene ablation during embryogenesis" to "... about the function of Dyrk2 during embryogenesis."

On line 115-116, the authors conclude that Dyrk2 is indispensable for survival during embryogenesis; however, their results show that the embryos survive until birth in the expected ratio but then die shortly after birth. These findings suggest that Dyrk2 is required for survival after birth and that it likely plays a role in embryonic organ development as they go on to show in the subsequent sections.

In the gene expression section starting on line 171, could the authors include a statement that these gene expression studies were conducted using RNA collected from whole embryos. This is clear in the methods but it is helpful to include in the results here because they have described multiple structural malformations in a wide range of organs but here they are describing global changes in gene expression.

In the results section describing the embryonic lung phenotypes the authors conclude that Dyrk2 is necessary for the development of both airway and alveolar branching (line 247-248). The branching defects that they are describing in this section are all airway branching. Alveolar development (septation) does not begin until after birth in mice and the authors have not investigated this process. Changing this conclusion to something like "These data demonstrate that Dyrk2 is necessary for airway branching" would be better.

The embryonic and cell type-specific expression of Dyrk2 is still unclear but much better. At E11, when the authors suggest that loss of Dyrk2 function is having a significant impact on Foxf1 expression, the epithelial expression of Dyrk2 is not clearly demonstrated; however the lack of Dyrk2 expression in the mutant embryos is clear at this stage.

The in vitro rescue with SAG in Dyrk2 mutant lung culture experiments helped strengthen the argument that loss of Dyrk2 has an impact on the Shh/Foxf1 pathway

The findings regarding reduced proliferation and apoptosis in lines 275-276 should come earlier in this section when the authors are describing the lung hypoplasia phenotype. Can the authors add some context to this finding?

In the discussion the authors make several interesting points. One that they could expand on is how Dyrk2 appears to regulate Shh/Foxf1 in lung development but that Shh and Foxf1 null embryos have more severe phenotypes. Could the authors expand on this - why does loss of an apparent upstream regulator have a weaker/less severe phenotype than loss of the downstream factors?

In the figure legends - a general comment is that the authors are only labelling the images. It would be helpful to include brief statements regarding the results of the figures so that the figures and legends can stand independent of the text in the manuscript.

The authors have made several changes to the figures that are very helpful. Please make sure that all of the panels are correctly labelled in the legends. For example in the legend for figure 1a,b there are lateral views of whole embryos, H&E sections (a) and skeleton preps (b)? There are some differences between the legend that is included with the figure and the figure legends included in the text of the manuscript. For example also in figure 1 there is a difference in the legend text in the manuscript where the gross morphology of the kidneys (should be "l" not "k") and lungs (should be "m" not "l") is presented.

Finally, in figure 3 the authors could zoom in or crop the micro-CT images to more clearly show the difference in lung structure/inflation.

Thank you very much again for the chance to review this much improved manuscript.

REVIEWERS' COMMENTS:

Reviewer #2 (Remarks to the Author):

Thank you very much again for the chance to review the manuscript "Dyrk2-deficiency exhibits congenital malformations with lung hypoplasia by impaired Shh-Foxf1 axis". The authors have done a very good job of addressing the prior review and their manuscript is significantly improved.

The authors have demonstrated that Dyrk2 plays an important role in the development of multiple organs in mice and that, in the lung, Dyrk2 plays an important role in regulating an essential Shh/Foxf1 molecular pathway.

My suggestions are minor and can be addressed by word editing only.

The manuscript is well written and generally the text is clear; however, there are many minor tense or singular/plural mistakes that can be addressed by editing.

In the introduction there is a typo on line 69 ("format" instead of "format").

In the results on line 103 the authors might consider changing the text from "little is known about the impact of Dyrk2 gene ablation during embryogenesis" to "... about the function of Dyrk2 during embryogenesis."

On line 115-116, the authors conclude that Dyrk2 is indispensable for survival during embryogenesis; however, their results show that the embryos survive until birth in the expected ratio but then die shortly after birth. These findings suggest that Dyrk2 is required for survival after birth and that it likely plays a role in embryonic organ development as they go on to show in the subsequent sections.

In the gene expression section starting on line 171, could the authors include a statement that these gene expression studies were conducted using RNA collected from whole embryos. This is clear in the methods but it is helpful to include in the results here because they have described multiple structural malformations in a wide range of organs but here they are describing global changes in gene expression.

In the results section describing the embryonic lung phenotypes the authors conclude that Dyrk2 is necessary for the development of both airway and alveolar branching (line 247-

248). The branching defects that they are describing in this section are all airway branching. Alveolar development (septation) does not begin until after birth in mice and the authors have not investigated this process. Changing this conclusion to something like "These data demonstrate that Dyrk2 is necessary for airway branching" would be better.

The embryonic and cell type-specific expression of Dyrk2 is still unclear but much better. At E11, when the authors suggest that loss of Dyrk2 function is having a significant impact on Foxf1 expression, the epithelial expression of Dyrk2 is not clearly demonstrated; however the lack of Dyrk2 expression in the mutant embryos is clear at this stage.

The in vitro rescue with SAG in Dyrk2 mutant lung culture experiments helped strengthen the argument that loss of Dyrk2 has an impact on the Shh/Foxf1 pathway

The findings regarding reduced proliferation and apoptosis in lines 275-276 should come earlier in this section when the authors are describing the lung hypoplasia phenotype. Can the authors add some context to this finding?

In the discussion the authors make several interesting points. One that they could expand on is how Dyrk2 appears to regulate Shh/Foxf1 in lung development but that Shh and Foxf1 null embryos have more severe phenotypes. Could the authors expand on this - why does loss of an apparent upstream regulator have a weaker/less severe phenotype than loss of the downstream factors?

In the figure legends - a general comment is that the authors are only labelling the images. It would be helpful to include brief statements regarding the results of the figures so that the figures and legends can stand independent of the text in the manuscript.

The authors have made several changes to the figures that are very helpful. Please make sure that all of the panels are correctly labelled in the legends. For example in the legend for figure 1a,b there are lateral views of whole embryos, H&E sections (a) and skeleton preps (b)? There are some differences between the legend that is included with the figure and the figure legends included in the text of the manuscript. For example also in figure 1 there is a difference in the legend text in the manuscript where the gross morphology of the kidneys (should be "l" not "k") and lungs (should be "m" not "l") is presented.

Finally, in figure 3 the authors could zoom in or crop the micro-CT images to more clearly show the difference in lung structure/inflation.

Thank you very much again for the chance to review this much improved manuscript.

Point-by-point response to the reviewers:

Thanks for your comments. We appreciate your effort and time. We have carefully considered each point of criticism and suggestion to provide a point-by-point response below. In this revision, the text in blue represents our response to your comments.

REVIEWERS' COMMENTS:

Reviewer #2 (Remarks to the Author):

Thank you very much again for the chance to review the manuscript "Dyrk2-deficiency exhibits congenital malformations with lung hypoplasia by impaired Shh-Foxf1 axis". The authors have done a very good job of addressing the prior review and their manuscript is significantly improved.

The authors have demonstrated that Dyrk2 plays an important role in the development of multiple organs in mice and that, in the lung, Dyrk2 plays an important role in regulating an essential Shh/Foxf1 molecular pathway.

My suggestions are minor and can be addressed by word editing only.

The manuscript is well written and generally the text is clear; however, there are many minor tense or singular/plural mistakes that can be addressed by editing.

[Response]

Thank you for the suggestion. We corrected the text.

In the introduction there is a typo on line 69 ("format" instead of "format").

[Response]

As suggested, we corrected a typo "format" (line 69).

In the results on line 103 the authors might consider changing the text from "little is known about the impact of Dyrk2 gene ablation during embryogenesis" to "... about the function of Dyrk2 during embryogenesis."

[Response]

As suggested, we corrected the text (line 102).

On line 115-116, the authors conclude that Dyrk2 is indispensable for survival during embryogenesis; however, their results show that the embryos survive until birth in the expected ratio but then die shortly after birth. These findings suggest that Dyrk2 is required for survival after birth and that it likely plays a role in embryonic organ development as they go on to show in the subsequent sections.

[Response]

This is important comment. We corrected the text as follows:

“These findings indicate that Dyrk2 is required for survival after birth and that it likely plays a role in embryonic organ development. (line 115-116)”

In the gene expression section starting on line 171, could the authors include a statement that these gene expression studies were conducted using RNA collected from whole embryos. This is clear in the methods but it is helpful to include in the results here because they have described multiple structural malformations in a wide range of organs but here they are describing global changes in gene expression.

[Response]

Thank you for the suggestion. We revised this sentence to “we compared gene expression profiles between the WT and Dyrk2^{-/-} embryos at E8.5 and E10.5 using RNA collected from whole embryos. (line 178-179)”

In the results section describing the embryonic lung phenotypes the authors conclude that Dyrk2 is necessary for the development of both airway and alveolar branching (line 247-248). The branching defects that they are describing in this section are all airway branching. Alveolar development (septation) does not begin until after birth in mice and the authors have not investigated this process. Changing this conclusion to something like "These data demonstrate that Dyrk2 is necessary for airway branching" would be better.

[Response]

Thank you for the valuable comment. We agree with the reviewer and as suggested, we revised this sentence to “These observations suggest that Dyrk2 is necessary for the development of airway branching. (line 254-255)”

The embryonic and cell type-specific expression of Dyrk2 is still unclear but much better. At E11, when the authors suggest that loss of Dyrk2 function is having a significant impact on Foxf1 expression, the epithelial expression of Dyrk2 is not clearly demonstrated; however the lack of Dyrk2 expression in the mutant embryos is clear at this stage.

[Response]

Thank you for the comment.

The in vitro rescue with SAG in Dyrk2 mutant lung culture experiments helped strengthen the argument that loss of Dyrk2 has an impact on the Shh/Foxf1 pathway

[Response]

Thank you for the comment.

The findings regarding reduced proliferation and apoptosis in lines 275-276 should come earlier in this section when the authors are describing the lung hypoplasia phenotype. Can the authors add some context to this finding?

[Response]

Thank you for the valuable comment. As suggested, we corrected Fig. 4k to new Fig. 5c and added to the text as follows:

“Furthermore, while the normal lung showed abundant cell proliferation and cell death in mesenchyme, the *Dyrk2*^{-/-} lungs reduced both cell proliferation and cell death (Figure 5c). (line 252-254)”

In the discussion the authors make several interesting points. One that they could expand

on is how *Dyrk2* appears to regulate *Shh*/*Foxf1* in lung development but that *Shh* and *Foxf1* null embryos have more severe phenotypes. Could the authors expand on this - why does loss of an apparent upstream regulator have a weaker/less severe phenotype than loss of the downstream factors?

[Response]

Thank you for the suggestion. As mentioned, *Dyrk2* acts as an upstream regulator of the *Shh*-*Foxf1* signaling and *Dyrk2*^{-/-} mice display mild phenotypes than *Shh*^{-/-} and *Foxf1*^{-/-} mice. Since, however, *Shh* and *Foxf1* expression were significantly reduced but not completely abolished in *Dyrk2*^{-/-} mice, abnormal phenotypes in *Dyrk2*^{-/-} mice were not severe than those in *Shh*^{-/-} and *Foxf1*^{-/-} mice.

We added this point in the Discussion (line 353-355).

In the figure legends - a general comment is that the authors are only labelling the images. It would be helpful to include brief statements regarding the results of the figures so that the figures and legends can stand independent of the text in the manuscript.

[Response]

Thank you for the suggestion. We added brief statements regarding the results of the figures in the figure legends (line 754-755, 769-770, 799, 813-814, 828-829).

The authors have made several changes to the figures that are very helpful. Please make sure that all of the panels are correctly labelled in the legends. For example in the legend for figure 1a,b there are lateral views of whole embryos, H&E sections (a) and skeleton preps (b)? There are some differences between the legend that is included with the figure and the figure legends included in the text of the manuscript. For example also in figure 1 there is a difference in the legend text in the manuscript where the gross morphology of the kidneys (should be "l" not "k") and lungs (should be "m" not "l") is presented.

[Response]

Thank you for the suggestion. We corrected the figure legends and panel labels.

Finally, in figure 3 the authors could zoom in or crop the micro-CT images to more clearly

show the difference in lung structure/inflation.

[Response]

Thank you for the comment. As suggested, we corrected the new Figure 4b and Figure S9.

New Fig. 4b

New Fig. S9